# Flavor deformations and supersymmetry enhancement
# in $4d$ $\mathcal{N} = 2$ theories

**Usman Naseer[★] and Charles Thull[†]**

Department of Physics and Astronomy, Uppsala University,
Box 516, SE-751 20 Uppsala, Sweden

★ usman.naseer@physics.uu.se, † charles.thull@physics.uu.se

## Abstract

We study $\mathcal{N} = 2$ theories on four-dimensional manifolds that admit a Killing vector $v$ with isolated fixed points. It is possible to deform these theories by coupling position-dependent background fields to the flavor current multiplet. The partition function of the deformed theory only depends on the value of the background scalar fields at the fixed points of $v$. For a single adjoint hypermultiplet, the partition function becomes independent of the supergravity as well as the flavor background if the scalars attain special values at the fixed points. For these special values, supersymmetry at the fixed points enhances from the Donaldson-Witten twist to the Marcus twist or the Vafa-Witten twist of $\mathcal{N} = 4$ SYM. Our results explain the recently observed squashing independence of $\mathcal{N} = 2^*$ theory on the squashed sphere and provide a new squashing independent point. Interpreted through the AGT-correspondence, this implies the $b$-independence of torus one-point functions of certain *local* operators in Liouville/Toda CFT. The position-dependent deformations imply relations between correlators of partially integrated operators in *any* $\mathcal{N} = 2$ SCFT with flavor symmetries.



# 1  Introduction and summary

A fruitful way to study a QFT is to introduce position-dependent couplings for various operators in the theory. The theory has some enhanced symmetry when couplings vanish[1]. Non-zero couplings break the symmetry. Position-dependent couplings can restore the enhanced symmetry if we assign to them judicious transformation rules under the symmetry transformations. Observables of the theory obtained after path-integrating over the dynamical fields must then be consistent with the enhanced symmetry. This idea has been used to obtain powerful results in QFT, for example, the non-perturbative $\beta$-function [1], non-renormalization theorems [2] in supersymmetric gauge theories and constraints on the RG flow in generic $4d$ quantum field theories [3,4].

Since every QFT has a stress tensor, one can always deform by coupling the stress tensor to a background metric. For generic QFTs the partition function is a scheme-dependent, non-local functional of the background metric and can only be computed in very special circumstances. The situation improves for supersymmetric theories. The supersymmetric Lagrangian with a background metric can be obtained by taking the rigid limit of an appropriate supergravity theory [5]. This corresponds to deforming the flat space theory by coupling operators in the stress tensor multiplet to the background fields. Moreover, the restrictions on supersymmetric counter terms make the finite part of the free energy meaningful [6,7]. This program, starting from the seminal work of [8], has led to a great deal of insights into the dynamics of strongly coupled theories (see [9] for a nice review of these developments).

In this paper we focus on $4d$ $\mathcal{N}=2$ theories. Supersymmetric background and partition function for these theories were studied in [10–12]. It was shown that the partition function depends on the supersymmetric background only through the Killing vector $v$ which is a bilinear of Killing spinors. To be explicit, $v$ has fixed points where either the anti-chiral or the chiral spinor vanishes — referred to as *plus* and *minus* fixed points respectively. Near its fixed points, $v$ defines a $T^2$-action with two parameters and the partition function only depends on

---

[1]Or are tuned to some special value.

these so-called equivariant parameters.

$$\mathcal{Z} = \mathcal{Z}\left(\{\tau_i\}, \{\varepsilon_x, \varepsilon_x'\}, \{\varepsilon_y, \varepsilon_y'\}\right), \tag{1.1}$$

where $\tau_i$ is the complexified coupling constant for each semi-simple factor in the gauge group, $\left(\varepsilon_x, \varepsilon_x'\right)$ are the equivariant parameters around the *plus* fixed points $x$ and $\left(\varepsilon_y, \varepsilon_y'\right)$ are the equivariant parameters around the *minus* fixed points $y$.

In the presence of matter, the theory has a flavor symmetry $F$. We study the flavor deformations of $\mathcal{N} = 2$ theories on the curved space by introducing position-dependent background fields for $F$. This can be done in a supersymmetric way by coupling the flavor current multiplet with a background vector multiplet. We show that the supersymmetry requires all the background fields to be constant along $v$. Moreover, the background gauge and auxiliary fields are determined in terms of the background scalars $m_a$ and $\overline{m}_a$ where $a = 1, 2, \cdots, \text{rank}(F)$. The partition function of the deformed theory only depends on the value of the scalars at the fixed points of $v$,

$$\mathcal{Z} = \mathcal{Z}\left(\{\tau_i\}, \{\varepsilon_x\}, \{\varepsilon_y\}, \{m_{a,x}\}, \{\overline{m}_{a,y}\}\right). \tag{1.2}$$

For constant $m_a = \overline{m}_a$, this reduces to turning on the usual mass terms for matter multiplets. Smooth deformations of $m_a$ and $\overline{m}_a$, which do not change their values at the fixed points, do not change the partition function.

A remarkable situation arises if the background couplings for various operators are correlated in such a way that the generating functional is independent of some of them. It was found in [7,13] that the free energy of certain mass-deformed theories on the squashed sphere $\mathbb{S}_b^d$ ($d = 3, 4$) becomes squashing-independent when the mass is tuned to $\pm\frac{i}{2}\left(b - b^{-1}\right)$. Here we find a remarkable generalization of this result for the flavor deformed $\mathcal{N} = 4$ theory on any curved $4d$ background. We show that if $m_x = \pm\frac{i}{2}\left(\varepsilon_x - \varepsilon_x'\right)$ and $\overline{m}_y = \pm\frac{i}{2}\left(\varepsilon_y + \varepsilon_y'\right)$ then the perturbative and non-perturbative contributions simplify at each fixed point and the partition function becomes independent of the equivariant parameters for the supergravity background. We also show that a similar simplification happens for $m_x = \pm\frac{i}{2}\left(\varepsilon_x + \varepsilon_x'\right)$ and $\overline{m}_y = \pm\frac{i}{2}\left(\varepsilon_y - \varepsilon_y'\right)$.

We are then naturally led to search for the underlying mechanism for this remarkable simplification. For squashed three-sphere, the relevant study was done in [14]. It was shown, based on earlier results [15,16], that the supersymmetry enhancement at the special values of the masses leads to the simplification of the free energy. Since the supersymmetric partition function of $4d$ $\mathcal{N} = 2$ theories only receives contributions from the fixed points of $v$ it suffices to focus on the structure of the theory near the fixed points. The general theory near a plus (minus) fixed point corresponds to the Donaldson-Witten [17] or half twist of the $\mathcal{N} = 4$ SYM which has two supercharges of positive (negative) chirality. We show that for special values of the masses the supersymmetry enhances. For the first set of masses, the theory around the fixed point corresponds to the Marcus twist [18] of $\mathcal{N} = 4$ SYM which has two additional supercharges of negative (positive) chirality. For the second simplification, the theory around the fixed point corresponds to the Vafa-Witten twist [19] of the $\mathcal{N} = 4$ SYM which has two additional supercharges of positive (negative) chirality.

After understanding the simplification at special masses we turn to several interesting applications of our results. First, these explain the recently observed squashing independence of the free energy of $\mathcal{N} = 2^*$ and also provide another squashing independent point when the mass is tuned to $\pm\frac{1}{2}\left(b + b^{-1}\right)$. Second, using the AGT correspondence [20], our results imply that the torus one-point functions of certain *local* operators in the Liouville/Toda field theory are independent of $b$. These operators do not correspond to normalizable states in the spectrum of the theory. Finally, independence of the free energy from the profile of the background scalar can be leveraged to obtain an infinite number of relations between correlation functions

of partially integrated operators in *any* $\mathcal{N} = 2$ SCFT with flavor symmetries. This generalizes earlier results [7, 21–24] in the literature on relations between integrated correlators of $\mathcal{N} = 4$ SYM.

The rest of the paper is organized as follows. In section 2 we review general aspects of $\mathcal{N} = 2$ theories on curved space and study their localized partition functions. In section 3 we introduce the position dependent deformation for the flavor current multiplet and show that the partition function only depends on the values of the background scalar at the fixed point of $v$. Specializing to the $\mathcal{N} = 2^*$ theory, we show that for special values of the background scalar the free energy becomes independent of the equivariant parameters. We then show in section 4 that the reason for this simplification is the symmetry enhancement that happens near the fixed points of $v$ for special masses. Finally we discuss various applications of our results in section 5.

# 2 $\mathcal{N} = 2$ supersymmetric theories on curved space

In this section we review the general aspects of $\mathcal{N} = 2$ supersymmetric theories on curved manifolds following [10, 12].

## 2.1 Lagrangians and background fields

An $\mathcal{N} = 2$ theory can be placed on a curved manifold by coupling it to the $\mathcal{N} = 2$ conformal supergravity and then taking the rigid limit à la [5]. The field content of the supergravity can conveniently be organized in terms of the gauge fields for the generators of the $\mathcal{N} = 2$ superconformal algebra. The gauge fields for translations, dilatations, $U(1)_R$ and $SU(2)_R$ R-symmetries, and $Q$-supersymmetry are [25]

$$g_{\mu\nu}, \qquad d_\mu, \qquad G_\mu, \qquad V^i_{\mu j}, \qquad \psi^i_\mu, \tag{2.1}$$

where $i, j = 1, 2$ are $SU(2)_R$ R-symmetry indices. The gauge fields for the Lorentz transformations, special conformal transformations and $S$-supersymmetries are not independent fields. The supergravity multiplet also has a set of auxiliary fields

$$B_{\mu\nu}, \qquad \chi^i, \qquad D, \qquad S_{(ij)}, \qquad \mathcal{F}_{\mu\nu}. \tag{2.2}$$

The supersymmetric backgrounds are then obtained by requiring that the fermionic fields $\psi^i_\mu, \chi^i$ and their supersymmetry variations vanish. These conditions are analyzed in detail in [10, 12] and explicit expressions for various background fields are obtained[2]. The supersymmetric backgrounds are obtained by requiring that

- The metric $g$ admits a smooth real Killing vector field $v$ with (at-most) isolated fixed points.

- There exist two smooth functions $s$ and $\tilde{s}$ that are constant along $v$, $g_{\mu\nu}v^\mu v^\nu = s\tilde{s}$, and $s + \tilde{s}$ approaches the same constant $K$ at every fixed point of $v$. We set $K = 1$ in this paper.

---

[2]The fields $B_{\mu\nu}$ and $D$ are related to $W_{\mu\nu}$ and N of [10, 12] by

$$W_{\mu\nu} = 8\sqrt{2}B_{\mu\nu}, \qquad \mathrm{N} = \frac{D}{2}. \tag{2.3}$$

All the background fields can then be obtained in terms of $g, v, s$ and $\widetilde{s}$, albeit not uniquely. The ambiguity is parameterized by two one-forms $G_\mu$ and $b_\mu$ such that $v^\mu b_\mu = 0$. We have listed all the expressions for the Killing spinors and the background fields in appendix A.

The Lagrangian of the $\mathcal{N} = 2$ vector multiplet on a supersymmetric background then takes the form

$$\mathcal{L}_{\text{vec}} = \mathcal{L}_{\text{vec}}^{\text{cov}} + \text{Tr}\Big[16B^{\mu\nu}(F^+_{\mu\nu}\overline{X} + F^-_{\mu\nu}X)) - 64B^{+\mu\nu}B_{+\mu\nu}\overline{X}^2 + \quad 64B^{-\mu\nu}B_{-\mu\nu}X^2$$
$$-2(D - \frac{R}{3})X\overline{X}\Big], \quad (2.4)$$

where $\mathcal{L}_{\text{vec}}^{\text{cov}}$ is obtained from the flat space Lagrangian by covariantizing all derivatives with respect to the metric and background gauge fields for the R-symmetry, and $X$ is the complex scalar field of the vector multiplet. Similarly for the hypermultiplet Lagrangian one needs to add non-minimal couplings with the background two-form, curvature and the scalar field to preserve supersymmetry,

$$\mathcal{L}_{\text{hyp}} = \mathcal{L}_{\text{hyp}}^{\text{cov}} + i B_{\mu\nu}\text{Tr}\Big(\psi_1\sigma^{\mu\nu}\psi_2 - \overline{\psi}_1\overline{\sigma}^{\mu\nu}\overline{\psi}_2\Big) - \frac{1}{4}\Big(D - \frac{2}{3}R\Big)\text{Tr}\Big(Z_1\overline{Z}_1 + Z_2\overline{Z}_2\Big), \quad (2.5)$$

where $\psi_i$ and $Z_i$ are fermions and scalars in the hypermultiplet.

The hypermultiplet action is $Q$-exact while the vector-multiplet action is $Q$-exact except at the fixed-points of the Killing vector $v$. Modulo $Q$-exact terms the action can be written as the sum of local operators at fixed points of $v$ [7, 26],

$$S = -4\pi i\tau \sum_{x:s(x)=0} \frac{\text{Tr}(X^2)(x)}{\varepsilon_x \varepsilon'_x} - 4\pi i\overline{\tau} \sum_{y:\widetilde{s}(y)=0} \frac{\text{Tr}(\overline{X}^2)(y)}{\varepsilon_y \varepsilon'_y}. \quad (2.6)$$

## 2.2 Localization and partition functions

Using localization, the partition function of the theory can be computed in terms of a matrix integral where the integrand is a product of classical, perturbative and non-perturbative contributions.

$$\mathcal{Z} = \int d^{r_G}\sigma \left(\prod_{\rho\in\Delta_+} |\rho(\sigma)|^2\right) Z_{\text{classical}} Z_{\text{Nek}} Z_{1-\text{loop}}^{\text{vec}} Z_{1-\text{loop}}^{\text{hyp}}. \quad (2.7)$$

$Z_{\text{classical}}$ is obtained by evaluating the action on the localization locus. The localization locus, in the absence of fluxes, is parameterized by the vector-multiplet scalar which takes a constant value $\sigma$ in the Cartan of the gauge group. To simplify notation we do not include fluxes in our explicit expressions here. The generalization to backgrounds which support non-trivial fluxes is given in appendix B.

Using eq. (2.6) we can compute the classical contribution to the partition function

$$\log Z_{\text{classical}} = -\frac{16\pi^2}{g_{\text{YM}}^2}\text{Tr}(\sigma^2)s_-, \quad (2.8)$$

where

$$s_- = \sum_{x:s(x)=0} \frac{1}{\varepsilon_x \varepsilon'_x} - \sum_{y:\widetilde{s}(y)=0} \frac{1}{\varepsilon_y \varepsilon'_y}. \quad (2.9)$$

$Z_{\text{Nek}}$ is the Nekrasov partition function which captures the contribution of localized instantons or anti-instantons at the fixed-points of $v$. Instantons contribute at the *plus* fixed points where $\widetilde{s} = 0$ and anti-instantons contribute at the *minus* fixed points where $s = 0$. Near

the fixed-points the theory coincides with the $\Omega$-deformed theory with equivariant parameters identified with the parameters of the $T^2$-action of $v$ [27]. We can express the non-perturbative contribution as

$$Z_{\text{Nek}} = \prod_{x:s(x)=0} Z^{\text{anti}-\text{inst}}_{\varepsilon_x,\varepsilon'_x}(i\sigma, \vec{m}, q) \prod_{y:\widetilde{s}(y)=0} Z^{\text{inst}}_{\varepsilon_y,\varepsilon'_y}(i\sigma, \vec{m}, q), \tag{2.10}$$

where $q = e^{2\pi i\tau}$, $\vec{m}$ is the vector of mass parameters for $\mathcal{N} = 2$ hypermultiplets, $Z^{\text{inst}}_{\varepsilon_x,\varepsilon'_x}$ and $Z^{\text{anti}-\text{inst}}_{\varepsilon_y,\varepsilon'_y}$ are equivariant instanton and anti-instanton partition functions in the $\Omega$-background [28, 29].

$Z^{\text{vec}}_{1-\text{loop}}$ and $Z^{\text{hyp}}_{1-\text{loop}}$ are the one-loop determinants associated with the vector-multiplet and hypermultiplets. These are computed from an index, expanded in a series and then translated into an infinite product. This procedure needs regularization. Expressions for hypermultiplet one-loop determinants with various regularizations are given in [12] and the same can be done for the vector-multiplet one-loop determinants. It is unclear if there exists a preferred regularization scheme on an arbitrary manifold [3]. To keep the expressions readable, we pick a regularization which is compatible with the $\mathbb{S}^4$ [8,27]. We have checked that, with appropriate modifications, our conclusions hold for any choice of regularization.

The one-loop determinant for the vector-multiplet can be written as a product over contributions from each fixed point.

$$Z^{\text{vec}}_{1-\text{loop}} = \prod_{x:s(x)=0} Z^{\text{vec}}_{\varepsilon_x,\varepsilon'_x}(\sigma) \prod_{y:\widetilde{s}(y)=0} Z^{\text{vec}}_{\varepsilon_y,\varepsilon'_y}(\sigma). \tag{2.11}$$

The contribution from each fixed point depends on the Coulomb branch parameter $\sigma$ and equivariant parameters. The contributions are given by

$$Z^{\text{vec}}_{\varepsilon_x,\varepsilon'_x}(\sigma) = \prod_{\rho\in\mathbf{adj}} \prod_{n_1,n_2\geq 0} \left(i\rho(\sigma) + (n_1+1)\varepsilon_x + (n_2+1)\varepsilon'_x\right)^{\frac{1}{2}} \prod_{\substack{m_1,m_2\geq 0 \\ (m_1,m_2)\neq(0,0)}} \left(i\rho(\sigma) + m_1\varepsilon_x + m_2\varepsilon'_x\right)^{\frac{1}{2}},$$

$$Z^{\text{vec}}_{\varepsilon_y,\varepsilon'_y}(\sigma) = \prod_{\rho\in\mathbf{adj}} \prod_{n_1,n_2\geq 0} \left(i\rho(\sigma) - (n_1+1)\varepsilon_y + (n_2+1)\varepsilon'_y\right)^{\frac{1}{2}} \prod_{\substack{m_1,m_2\geq 0 \\ (m_1,m_2)\neq(0,0)}} \left(i\rho(\sigma) - m_1\varepsilon_y + m_2\varepsilon'_y\right)^{\frac{1}{2}}.$$

$$\tag{2.12}$$

Similarly, the hypermultiplet contribution also factorizes into contributions from the fixed points of $v$.

$$Z^{\text{hyp}}_{1-\text{loop}} = \prod_{x:s(x)=0} Z^{\text{hyp}}_{\varepsilon_x,\varepsilon'_x}(\sigma, m) \prod_{y:\widetilde{s}(y)=0} Z^{\text{hyp}}_{\varepsilon_y,\varepsilon'_y}(\sigma, m), \tag{2.13}$$

where $m$ is the mass of the hypermultiplet. The one-loop determinant at each fixed point takes the form

$$Z^{\text{hyp}}_{\varepsilon_x,\varepsilon'_x}(\sigma, m) = \prod_{\rho\in\mathbf{R}} \prod_{n_1,n_2\geq 0} \left(i\rho(\sigma) + im + (n_1+\tfrac{1}{2})\varepsilon_x + (n_2+\tfrac{1}{2})\varepsilon'_x\right)^{-\frac{1}{2}}$$

$$\times \left(-i\rho(\sigma) - im + \left(n_1+\tfrac{1}{2}\right)\varepsilon_x + \left(n_2+\tfrac{1}{2}\right)\varepsilon'_x\right)^{-\frac{1}{2}},$$

$$Z^{\text{hyp}}_{\varepsilon_y,\varepsilon'_y}(\sigma, m) = \prod_{\rho\in\mathbf{R}} \prod_{n_1,n_2\geq 0} \left(i\rho(\sigma) + im - \left(n_1+\tfrac{1}{2}\right)\varepsilon_y + (n_2+\tfrac{1}{2})\varepsilon'_y\right)^{-\frac{1}{2}}$$

$$\times \left(-i\rho(\sigma) - im - \left(n_1+\tfrac{1}{2}\right)\varepsilon_y + \left(n_2+\tfrac{1}{2}\right)\varepsilon'_y\right)^{-\frac{1}{2}}. \tag{2.14}$$

---

[3]We thank L. Ruggeri and R. Mauch for discussions on this point.

We now simplify the vector multiplet one-loop determinant. The product over $\rho$ contains a $\sigma$-independent contribution coming from each zero weight $\rho(\sigma) = 0$. This contribution is given in terms of

$$Z_{\varepsilon_x,\varepsilon_x'}^{U(1)\text{vec}} = \prod_{n_1,n_2 \geq 0} \left((n_1+1)\varepsilon_x + n_2\varepsilon_x'\right)^{\frac{1}{2}} \left(n_1\varepsilon_x + (n_2+1)\varepsilon_x'\right)^{\frac{1}{2}}, \qquad (2.15)$$

where $Z_{\varepsilon_x,\varepsilon_x'}^{U(1)\text{vec}}$ is the one-loop determinant for a free vector multiplet. The rest of the contribution is given by a product of $\rho$ over the positive roots of the Lie algebra.

$$\begin{aligned}
Z_{\varepsilon_x,\varepsilon_x'}^{\text{vec}} &= \left(Z_{\varepsilon_x,\varepsilon_x'}^{U(1)\text{vec}}\right)^{r_G} \prod_{\rho \in \Delta_+} \frac{1}{|\rho(\sigma)|} \prod_{n_1,n_2 \geq 0} \left(\rho(\sigma)^2 + \left((n_1+1)\varepsilon_x + (n_2+1)\varepsilon_x'\right)^2\right)^{\frac{1}{2}} \\
&\qquad \times \left(\rho(\sigma)^2 + \left(n_1\varepsilon_x + n_2\varepsilon_x'\right)^2\right)^{\frac{1}{2}}, \\
Z_{\varepsilon_y,\varepsilon_y'}^{\text{vec}} &= \left(Z_{-\varepsilon_y,\varepsilon_y'}^{U(1)\text{vec}}\right)^{r_G} \prod_{\rho \in \Delta_+} \frac{1}{|\rho(\sigma)|} \prod_{n_1,n_2 \geq 0} \left(\rho(\sigma)^2 + \left(-(n_1+1)\varepsilon_y + (n_2+1)\varepsilon_y'\right)^2\right)^{\frac{1}{2}} \\
&\qquad \times \left(\rho(\sigma)^2 + \left(-n_1\varepsilon_y + n_2\varepsilon_y'\right)^2\right)^{\frac{1}{2}}.
\end{aligned} \qquad (2.16)$$

We can also rewrite the hypermultiplet determinants by splitting the product over $\rho \in \mathbf{R}$. For $\rho$ in the zero-weight space of the representation $\mathbf{R}$ we get a $\sigma$-independent contribution which can be written in terms of

$$Z_{\varepsilon_x,\varepsilon_x'}^{U(1)\text{hyp}}(m) = {\prod_{n_1,n_2 \geq 0}}' \left(m^2 + \left(n_1\varepsilon_x + n_2\varepsilon_x' + \frac{\varepsilon_x + \varepsilon_x'}{2}\right)^2\right)^{-\frac{1}{2}}, \qquad (2.17)$$

where $\prod'$ denotes that we have omitted the term $a_{n_1,n_2}$ from the product if $a_{n_1,n_2} = 0$. Combining with the rest of the contribution, we find

$$\begin{aligned}
Z_{\varepsilon_x,\varepsilon_x'}^{\text{hyp}}(\sigma,m) &= \left(Z_{\varepsilon_x,\varepsilon_x'}^{U(1)\text{hyp}}(m)\right)^{|\mathbf{R}_0|} \prod_{\rho \in \mathbf{R}\backslash\mathbf{R}_0} \prod_{n_1,n_2 \geq 0} \left((\rho(\sigma)+m)^2 + \left(n_1\varepsilon_x + n_2\varepsilon_x' + \frac{\varepsilon_x + \varepsilon_x'}{2}\right)^2\right)^{-\frac{1}{2}}, \\
Z_{\varepsilon_y,\varepsilon_y'}^{\text{hyp}}(\sigma,m) &= \left(Z_{-\varepsilon_y,\varepsilon_y'}^{U(1)\text{hyp}}(m)\right)^{|\mathbf{R}_0|} \prod_{\rho \in \mathbf{R}\backslash\mathbf{R}_0} \prod_{n_1,n_2 \geq 0} \left((\rho(\sigma)+m)^2 + \left(-n_1\varepsilon_y + n_2\varepsilon_y' + \frac{-\varepsilon_y + \varepsilon_y'}{2}\right)^2\right)^{-\frac{1}{2}},
\end{aligned} \qquad (2.18)$$

where $|\mathbf{R}_0|$ is the dimension of the zero weight space $\mathbf{R}_0$ of the representation $\mathbf{R}$.

## 3 Position-dependent flavor deformations

Supersymmetric theories can be placed on a curved space by coupling the stress tensor multiplet to the appropriate background fields and taking the rigid limit. This can be refined further by coupling conserved current multiplets, other than the stress tensor multiplet, to background fields. Indeed, this is how the mass terms can be introduced systematically for matter multiplets on a curved space. Generically one turns on a constant scalar, which plays the role of mass, in the background vector multiplet. To preserve the supersymmetry other fields in the background multiplet are fixed in terms of the mass to obtain the deformed theory. In this section, we generalize this and show that it is possible to turn on position-dependent flavor deformations while respecting the symmetries of the theory. We then study the effect of

these deformations on the supersymmetric partition functions and observe that for the mass-deformed $\mathcal{N} = 4$ SYM the partition functions simplify considerably if the deformations are tuned appropriately.

## 3.1 Background fields for flavor deformations

We restrict ourselves to deformations in the Cartan of the flavor symmetry. To this end it suffices to focus on a single conserved current multiplet with bosonic components $\left(j^\mu, \Sigma, \overline{\Sigma}, B^{ij}\right)$ and couple it to the background multiplet $\left(A_\mu, m, \overline{m}, D_{ij}, \lambda_i, \overline{\lambda}_i\right)$. The general deformation in the Cartan of the flavor symmetry is then just given by a rank($F$)-tuple of the $U(1)$-deformations.

To preserve the supersymmetry we require that all the fermions of the background multiplet $\lambda_i, \overline{\lambda}_i$ and their supersymmetry variations vanish. It is most convenient to use the cohomological formulation of $\mathcal{N} = 2$ theory which we review in appendix A.1. The components of the vector multiplet fermion can be decomposed into a scalar part $\eta$, a one-form part $\Psi_\mu$ and a two-form part $\chi_{\mu\nu}$. In terms of the cohomological fields these constraints read

$$0 = \delta\eta = L_v\varphi\,, \tag{3.1}$$

$$0 = \delta\Psi = \iota_v F + i\, d\phi\,, \tag{3.2}$$

$$0 = \delta\chi = H\,, \tag{3.3}$$

where $\varphi = -i(m - \overline{m})$ and $\phi = \tilde{s}m + s\overline{m}$ are two scalar combinations, $F = dA$ is the field strength of the background gauge field and $H$ is a two-form given in eq. (A.24). The last of these conditions fixes the auxiliary field $D_{ij}$ in terms of all the other background fields,

$$
\begin{aligned}
D_{ij} = 4\frac{s^2 + \tilde{s}^2}{(s + \tilde{s})^2}\widehat{\Theta}_{ij}^{\rho\lambda}\Bigg( & F_{\rho\lambda} - i\frac{m + \overline{m}}{s + \tilde{s}}(\partial_\rho v_\lambda - \partial_\lambda v_\rho) \\
& + \frac{2i}{s + \tilde{s}}\epsilon_{\rho\lambda\gamma}{}^\delta v^\gamma\left(\left(D_\delta - 2iG_\delta - i\frac{\tilde{s}}{s + \tilde{s}}b_\delta\right)m - \left(D_\delta + 2iG_\delta - i\frac{s}{s + \tilde{s}}b_\delta\right)\overline{m}\right)\Bigg)\,, \tag{3.4}
\end{aligned}
$$

where $\widehat{\Theta}_{ij}^{\rho\lambda}$ is a bilinear defined in eq. (A.28). Next, we combine the explicit expressions $\varphi = i(\overline{m} - m)$, $\phi = s\overline{m} + \tilde{s}m$ with the first two constraints and use that $s, \tilde{s}$ are constant along the flow of $v$ to see that $m$ and $\overline{m}$ are also constant along $v$. Similarly taking the exterior derivative of the second constraint one finds that $F$ is also constant along $v$.

Let us now analyze the second constraint in detail. Locally $F = dA$ so that we can rewrite the constraint as[4]

$$i\, d\phi = -L_v A + d(\iota_v A)\,. \tag{3.5}$$

Assuming $A$ is constant along $v$, we can integrate the above equation to get

$$\iota_v A = i(\phi - c)\,. \tag{3.6}$$

for a constant $c$. If $A$ stays finite at a fixed point $x_*$ of $v$ then we necessarily have $c = \phi(x_*)$. A solution for the background gauge field is then

$$A = +i\frac{\phi - c}{s\tilde{s}}v\,. \tag{3.7}$$

This is constant along $v$. This expression for $A$ is valid in a patch which contains only the fixed point $x_*$. In a patch containing some other fixed point the above expression might not be well defined. Transitioning between patches we can however do a gauge transformation,

---

[4]We use that for every vector field $X$ and every differential form $\omega$ the relation $L_X\omega = \iota_X d\omega + d\iota_X\omega$ holds.

$A \to A + \mathrm{d}\Lambda$. In (3.6) this adds a term $\iota_v \mathrm{d}\Lambda$. As $\phi$ is gauge invariant this corresponds to a change in $c$,

$$\Delta c = \mathrm{i}\iota_v \mathrm{d}\Lambda. \tag{3.8}$$

Away from the zeros of $v$ this equation can be integrated so that arbitrary values of $c$ are possible in different patches. Specifically this means that for every $\phi$ we choose we can find a finite gauge field configuration that satisfies our constraint equations. We conclude that it is possible to introduce position-dependent flavor deformations where all background fields are constant along $v$ and the background gauge and auxiliary fields are determined in terms of the background scalar field. Moreover, as we show in appendix C, there is enough freedom to tune the background scalar at every fixed point of $v$ to whatever value we desire.

## 3.2 Partition function with a position-dependent deformation

We now discuss the partition functions for theories with a position-dependent flavor deformation. When the hypermultiplet couples to a background vector multiplet the relevant combination that appears in the one-loop determinants is [12,30]

$$\mathrm{i}\iota_v A + \phi \tag{3.9}$$

evaluated at the fixed points. This adds a shift to the term $\rho(\sigma)$ that appears in the one-loop determinants in eqs. (2.14) and (2.18). Since $A$ is finite, this combination evaluates to[5] $m_x$ at the *plus* fixed point $x$ and to $\overline{m}_y$ at the *minus* fixed point $y$. The one-loop determinants for the hypermultiplet at a fixed point are thus the ones in eqs. (2.14) and (2.18) with the constant $m$ now replaced by $m_x$ and $\overline{m}_y$ at the fixed points $x$ and $y$ respectively.

If the hypermultiplet transforms in the representation $\mathbf{R}_F$ of the flavor symmetry group then the complete one-loop determinant picks up a product over the weights $\rho_F$ of $\mathbf{R}_F$ with $m_x$ and $\overline{m}_y$ replaced by $\rho_F(m_x)$ or $\rho_F(\overline{m}_y)$.

## 3.3 $\mathcal{N} = 2^*$ with special masses at the fixed points

We now specialize to the flavor deformed theory with a single adjoint hypermultiplet. This is the $\mathcal{N} = 2^*$ theory with a position-dependent mass. Using our results from the previous section we show that when the mass is tuned to special values at the fixed points of $v$, the perturbative and the non-perturbative contributions to the partition function simplify considerably. In the next section we investigate the underlying reason for this simplification.

For the adjoint hypermultiplet we can split the one-loop determinant in terms of a product over the positive roots which takes the form

$$
\begin{aligned}
Z^{\mathrm{hyp}}_{\varepsilon_x,\varepsilon'_x}(\sigma, m_x) = {} & \left( Z^{U(1)\mathrm{hyp}}_{\varepsilon_x,\varepsilon'_x}(m_x) \right)^{r_G} \prod_{\rho \in \Delta_+} \prod_{n_1,n_2 \geq 0} \left( \rho(\sigma)^2 + \left( n_1\varepsilon_x + n_2\varepsilon'_x + \frac{\varepsilon_x + \varepsilon'_x}{2} + \mathrm{i}m_x \right)^2 \right)^{-\frac{1}{2}} \\
& \times \left( \rho(\sigma)^2 + \left( n_1\varepsilon_x + n_2\varepsilon'_x + \frac{\varepsilon_x + \varepsilon'_x}{2} - \mathrm{i}m_x \right)^2 \right)^{-\frac{1}{2}}, \\
Z^{\mathrm{hyp}}_{\varepsilon_y,\varepsilon'_y}(\sigma, \overline{m}_y) = {} & \left( Z^{U(1)\mathrm{hyp}}_{-\varepsilon_y,\varepsilon'_y}(\overline{m}_y) \right)^{r_G} \prod_{\rho \in \Delta_+} \prod_{n_1,n_2 \geq 0} \left( \rho(\sigma)^2 + \left( -n_1\varepsilon_y + n_2\varepsilon'_y + \frac{-\varepsilon_y + \varepsilon'_y}{2} + \mathrm{i}\overline{m}_y \right)^2 \right)^{-\frac{1}{2}} \\
& \times \left( \rho(\sigma)^2 + \left( -n_1\varepsilon_y + n_2\varepsilon'_y + \frac{-\varepsilon_y + \varepsilon'_y}{2} - \mathrm{i}\overline{m}_y \right)^2 \right)^{-\frac{1}{2}}.
\end{aligned}
\tag{3.10}
$$

---

[5]We use subscripts to denote value of $m$ and $\overline{m}$ at the fixed points of $v$.

We now explore the simplifications that happen when $m_x$ and $\overline{m}_y$ are tuned to special values.

### 3.3.1 The Marcus point

Let us now focus on the *plus* fixed point and tune the hypermultiplet mass to be

$$m_x = \pm m^*_{\varepsilon_x, -\varepsilon'_x} \equiv \pm i \frac{\varepsilon_x - \varepsilon'_x}{2}. \tag{3.11}$$

We refer to this as the Marcus point because, as we show in section 4, the theory near the fixed point corresponds to the Marcus twist of the $\mathcal{N} = 4$ SYM. For these values, the one-loop determinants of vector multiplets cancel exactly against the one-loop determinants of the hypermultiplet. Moreover the instanton contribution also becomes trivial [18] for any gauge group hence we have that

$$Z_{\text{Nek}} Z^{\text{vec}}_{1-\text{loop}} Z^{\text{hyp}}_{1-\text{loop}} = 1, \tag{3.12}$$

and the partition function becomes

$$\mathcal{Z} = \int d^{r_G} \sigma \left( \prod_{\rho \in \Delta_+} |\rho(\sigma)|^2 \right) \exp\left( -\frac{16\pi^2 s_-}{g_{\text{YM}}^2} \text{Tr}(\sigma^2) \right). \tag{3.13}$$

For *any* gauge group $G$ this matrix model can be easily solved giving the partition function

$$\mathcal{Z} = \tau_2^{-\frac{|G|}{2}}, \tag{3.14}$$

up to a coupling-independent overall constant and $|G|$ is the dimension of the gauge group. This is precisely the partition function of the theory on the round $\mathbb{S}^4$. It is interesting to analyze the behavior of the partition function under the duality transformations. While it is invariant under the T-duality it is not invariant under the S-duality. In fact $\log \mathcal{Z}$ is proportional to the Kähler potential and is not a single-valued function on the conformal manifold. It can, however, be made single-valued on the *extended* conformal manifold by introducing couplings to counter-terms [31]. This is equivalent to restoring the modular invariance of the partition function by adding a suitable counter-term.

### 3.3.2 The Vafa-Witten point

Another special value is when the mass at the *plus* fixed point is tuned to $\pm m^*_{\varepsilon_x, \varepsilon'_x}$ and at the *minus* fixed point it is tuned to $\pm m^*_{\varepsilon_y, -\varepsilon'_y}$. We call this the Vafa-Witten point because, as we show in section 4, the theory near the fixed point corresponds to the Vafa-Witten twist of $\mathcal{N} = 4$ SYM. A short calculation shows that the contribution of the vector multiplet and the adjoint hypermultiplet cancels almost entirely at that fixed point giving

$$Z^{\text{vec}}_{\varepsilon_x, \varepsilon'_x}(\sigma) Z^{\text{hyp}}_{\varepsilon_x, \varepsilon'_x}\left(\sigma, \pm m^*_{\varepsilon_x, \varepsilon'_x}\right) = \prod_{\rho \in \Delta_+} \frac{1}{|\rho(\sigma)|}. \tag{3.15}$$

The $k$-instanton contribution also becomes very simple and is equal to the Euler character of the moduli space of instantons on the $\Omega$-deformed flat space [8, 19, 32]. The full instanton contribution is then just the Vafa-Witten partition function on the $\Omega$-background which we denote by $Z^G_{\text{V.W}}$ for the gauge group $G$. For example, for the $U(N)$ gauge group the complete instanton partition function for this value of hypermultiplet mass becomes

$$Z^{U(N)}_{\text{V.W}}(\tau) = \prod_{n=1}^{\infty} \frac{1}{(1-q^n)^N} = \left( \frac{1}{q^{-\frac{1}{24}} \eta(\tau)} \right)^N. \tag{3.16}$$

Analogous simplifications occur at the *minus* fixed point if the mass is tuned to $\pm m^*_{-\varepsilon_y,\varepsilon'_y}$ giving

$$Z^{\text{vec}}_{\varepsilon_y,\varepsilon'_y}(\sigma)\, Z^{\text{hyp}}_{\varepsilon_y,\varepsilon'_y}\left(\sigma,\pm m^*_{-\varepsilon_y,\varepsilon'_y}\right) = \prod_{\rho\in\Delta_+}\frac{1}{|\rho(\sigma)|}, \tag{3.17}$$

and the non-perturbative contribution is simply $\overline{Z}^G_{\text{V.W}}(\overline{\tau}) \equiv \overline{Z^G_{\text{V.W}}(\tau)}$.

The full partition function for this special assignment of masses becomes

$$\mathcal{Z} = \left(Z^G_{\text{V.W}}\right)^{n_+}\left(\overline{Z}^G_{\text{V.W}}\right)^{n_-}\int d^{r_G}\sigma\,\exp\left(-\frac{16\pi^2 s_-}{g^2_{\text{YM}}}\text{Tr}(\sigma^2)\right)\prod_{\rho\in\Delta_+}|\rho(\sigma)|^{2-n_+-n_-}, \tag{3.18}$$

where $n_+(n_-)$ is the number of *plus* (*minus*) fixed points. It is interesting to see that the non-pertubative part above factorizes and the perturbative contribution is $\sigma$-dependent. The coupling dependence of the above matrix integral can be computed exactly and the partition function becomes

$$\mathcal{Z} = \left(Z^G_{\text{V.W}}\right)^{n_+}\left(\overline{Z}^G_{\text{V.W}}\right)^{n_-}\tau_2^{-\frac{|G|}{2}}\tau_2^{\frac{(n_++n_-)(|G|-r_G)}{4}}, \tag{3.19}$$

up to an overall coupling-independent constant.

Let us now analyze the behavior of the partition function under modular transformation. We specialize to $G = U(N)$.

$$\mathcal{Z} = \left(\frac{q^{\frac{1}{24}}}{\eta}\right)^{n_+ N}\left(\frac{\overline{q}^{\frac{1}{24}}}{\overline{\eta}}\right)^{n_- N}\tau_2^{-\frac{N^2}{2}}\tau_2^{\frac{N(N-1)(n_++n_-)}{4}}. \tag{3.20}$$

This is invariant under $\tau \to \tau + 1$ but not under $\tau \to -\frac{1}{\tau}$. On the topological sphere, $n_+ = n_- = 1$, one can use the Euler-density as a counter term [8] to obtain a modular invariant partition function $\left(\eta\overline{\eta}\sqrt{\tau_2}\right)^{-N}$. This can be generalized to the generic case. The relevant counter terms that can change the finite part of the partition function are (supersymmetrizations of) the Euler density, the Weyl-squared term, the Pontryagin density [7] and the mass-squared term[6] [33,34]. We do not consider the Weyl-squared term further as the partition function only depends on the topological data while the Weyl-squared term is not topological. Similarly, we do not consider the flavor background term further as the partition function is independent of the flavor background. We now rewrite the partition function as

$$\mathcal{Z} = \left(\frac{q^{\frac{1}{24}}\overline{q}^{\frac{1}{24}}}{\eta\overline{\eta}\sqrt{\tau_2}}\right)^{\frac{n_++n_-}{2}N}(\tau_2)^{\frac{N^2}{2}\left(\frac{n_++n_-}{2}-1\right)}\times\left[\frac{q^{\frac{1}{24}}\overline{\eta}}{\overline{q}^{\frac{1}{24}}\eta}\right]^{\frac{n_+-n_-}{2}N}, \tag{3.21}$$

separating the norm $|\mathcal{Z}|$ and a pure phase by the multiplication sign. We can make both $|\mathcal{Z}|$ and the phase modular invariant by using the Euler and Pontryagin-density counter terms respectively. The modular invariant partition function is then

$$\left(\eta\overline{\eta}\sqrt{\tau_2}\right)^{\frac{N(N-1)(n_++n_-)}{2}-N^2}. \tag{3.22}$$

# 4 Symmetry enhancement for $\mathcal{N} = 2^*$ near the fixed points

There are two key insights one obtains from the computation of the localized partition function using the index theorem. First, they factorize into contributions from the different fixed points

---

[6]We thank Yifan Wang for pointing out the supersymmetric flavor background counter-terms and their importance for the AGT correspondence.

of the Killing vector $v$, and second, these localized contributions only depend on the supersymmetry at the corresponding fixed points. It can be shown [27] that to first order around the fixed point the supersymmetry background approaches the $\Omega$-background[7] which is obtained by twisting the $R$-symmetry of the theory with an $SU(2)$ factor of the Lorentz group. For the $\mathcal{N} = 2^*$ theory with arbitrary mass parameter this twist corresponds to the Donaldson-Witten twist [17], also known as the half-twist of $\mathcal{N} = 4$. This twisted theory has two supercharges of the same chirality. We show that the special values of the mass correspond to the two other twists of the $\mathcal{N} = 4$ SYM each with two additional supercharges [18, 19, 35]. This supersymmetry enhancement helps explain the simplification of the free energy observed in previous sections. See also [36–38] for a discussion of mass-parameter and twists of $\mathcal{N} = 4$ SYM.

## 4.1   $\Omega$-deformation from dimensional reduction

In order to understand the symmetry enhancement we take the classical view of the $\Omega$-deformation [28]. We start with a 5$d$ theory which upon dimensional reduction with twisted boundary conditions gives the four-dimensional theory on the $\Omega$-background. We systematically analyze this construction to obtain the metric on the four-dimensional space as well as the generalized Killing spinor equations which are satisfied by the 4$d$ spinors corresponding to conserved supercharges.

The five dimensional geometry which upon dimensional reduction yields the $\Omega$-background with equivariant parameters $\varepsilon_1, \varepsilon_2$ is specified by

$$ds^2 = \delta_{\mu\nu}dx^\mu dx^\nu + d\vartheta^2, \tag{4.1}$$

with the identification

$$(x^\mu, \vartheta) \sim (R^\mu{}_\nu(\beta) x^\nu, \vartheta + \beta), \tag{4.2}$$

$\beta$ is the circumference of the compact direction but is otherwise an arbitrary constant. $R^\mu{}_\nu(\theta)$ is the matrix which rotates the $(x^1, x^2)$-plane by an angle $\varepsilon_1 \theta$ and the $(x^3, x^4)$-plane by an angle $\varepsilon_2 \theta$.

$$R^\mu{}_\nu(\theta) = \begin{pmatrix} \cos\varepsilon_1\theta & \sin\varepsilon_1\theta & 0 & 0 \\ -\sin\varepsilon_1\theta & \cos\varepsilon_1\theta & 0 & 0 \\ 0 & 0 & \cos\varepsilon_2\theta & \sin\varepsilon_2\theta \\ 0 & 0 & -\sin\varepsilon_2\theta & \cos\varepsilon_2\theta \end{pmatrix}. \tag{4.3}$$

We now do a coordinate change so that the four-dimensional space is independent of the periodicity conditions imposed on $\vartheta$. The periodicity conditions for the new set of coordinates $(x'^\mu, \vartheta) = (R^\mu{}_\nu(-\vartheta) x^\nu, \vartheta)$ are

$$(x'^\mu, \vartheta) \sim (x'^\mu, \vartheta + \beta). \tag{4.4}$$

The metric in the new coordinates becomes, after dropping the primes

$$ds^2 = (dx^\mu + v^\mu d\vartheta)^2 + (d\vartheta)^2, \tag{4.5}$$

where $v_\mu = \Omega_{\mu\nu} x^\nu$ and

$$\Omega = \varepsilon_1 dx^1 \wedge dx^2 + \varepsilon_2 dx^3 \wedge dx^4. \tag{4.6}$$

The non-zero components of the torsionless spin-connection in the obvious frame

$$E^a = dx^a + v^a d\vartheta, \qquad E^5 = d\vartheta, \tag{4.7}$$

---

[7] [27] analyzed the case of a topological sphere which has two fixed points of $v$ but the generalization to a manifold with many fixed points is straightforward

are

$$\omega_\vartheta{}^{ab} = \frac{1}{2} E^{a\mu} E^{b\nu} \left( \partial_\nu v_\mu - \partial_\mu v_\nu \right).$$
(4.8)

The indices $a, b = 1, 2, \cdots, 4$ are the local frame indices. It is possible to turn on a background field for the $SU(2)_R$ R-symmetry along the compact direction. The preserved Killing spinors then satisfy the 5$d$ generalized Killing spinor equations

$$\nabla_\mu \xi^i = 0, \qquad \nabla_\vartheta \xi^i + iA^i{}_j \xi^j = 0,$$
(4.9)

where $A^i{}_j$ is the component of the $SU(2)_R$ background field along the circle. The first condition is trivial to satisfy. Only the Killing spinors independent of $\vartheta$ are compatible with the dimensional reduction. Then, the second condition relates a Lorentz rotation of the spinors to an $R$-symmetry rotation

$$\frac{1}{4} \partial_\mu v_\nu \Gamma^{\mu\nu} \xi^i + iA^i{}_j \xi^j = 0.$$
(4.10)

In our conventions for the gamma matrices (given in appendix A.1), the matrix $\frac{1}{4} \partial_\mu v_\nu \Gamma^{\mu\nu}$ is a diagonal matrix with entries $\frac{i}{2} (\varepsilon_2 - \varepsilon_1, \varepsilon_1 - \varepsilon_2, -\varepsilon_1 - \varepsilon_2, \varepsilon_1 + \varepsilon_2)$. If now we turn on the Wilson line

$$A^i{}_j = \frac{1}{2} (\varepsilon_1 + \varepsilon_2)(\tau_3)^i{}_j$$
(4.11)

then the following spinors of positive chirality are preserved.

$$\xi_0^1 = (0, 0, 1, 0)^{\mathrm{T}}, \qquad \xi_0^2 = (0, 0, 0, 1)^{\mathrm{T}}.$$
(4.12)

This amounts to twisting the $SU(2)_r \subset Spin(4)$ with the R-symmetry group so that the new Lorentz group is $SU(2)_l \times [SU(2)_r \times SU(2)_R]_{\mathrm{diag}}$. This is the Donaldson-Witten twist of $\mathcal{N} = 2$ theories [17].

If the theory has a flavor symmetry $F$ then masses for the matter multiplets can be introduced by turning on the component $A_F$ of the flavor background field along the circle which takes values in the Cartan of $F$

$$A_F = i \sum_{i=1}^{F} m_i H_i,$$
(4.13)

where $H_i$ are the generators of the Cartan of F. This construction also holds for the partition function of the 4$d$ theory. The partition function is obtained from the $\beta \to 0$ limit of the index of the five dimensional theory with appropriate fugacities turned on [28].

Let us now specialize to $\mathcal{N} = 4$ SYM and show the existence of additional Killing spinors at special values of mass parameters. We decompose the representation $\mathbf{4}$ of $SU(4)_R$ into $U(1) \times SO(4)_R = U(1) \times SU(2)_R \times SU(2)_F$ as

$$\mathbf{4} = (\mathbf{2}, \mathbf{1})_{\frac{1}{2}} + (\mathbf{1}, \mathbf{2})_{-\frac{1}{2}}.$$
(4.14)

$(\xi^1, \xi^2)$ then transform as the $\mathbf{2}$ of $SU(2)_R \subset SO(4)_R$ which we identify with the $\mathcal{N} = 2$ R-symmetry and $(\xi^3, \xi^4)$ transform as the $\mathbf{2}$ of $SU(2)_F \subset SO(4)_R$ which we identify with the flavor symmetry of the theory. The Wilson line for the flavor symmetry is

$$A_F = im\tau_3.$$
(4.15)

In addition to $\xi^{1,2}$ the spinors, $\xi^{3,4}$ will be preserved if they are independent of the four-dimensional coordinates and satisfy

$$\frac{1}{4} \partial_\mu v_\nu \Gamma^{\mu\nu} \xi^{3,4} \mp m \xi^{3,4} = 0.$$
(4.16)

Table 1: The mass parameters and additional Killing spinors that solve the constraint eq. (4.16).

| $m$ | $\xi_0^3$ | $\xi_0^4$ |
|---|---|---|
| $+\frac{i}{2}(\varepsilon_1 - \varepsilon_2)$ | $(0,1,0,0)^{\mathrm{T}}$ | $(1,0,0,0)^{\mathrm{T}}$ |
| $-\frac{i}{2}(\varepsilon_1 - \varepsilon_2)$ | $(1,0,0,0)^{\mathrm{T}}$ | $(0,1,0,0)^{\mathrm{T}}$ |
| $+\frac{i}{2}(\varepsilon_1 + \varepsilon_2)$ | $(0,0,0,1)^{\mathrm{T}}$ | $(0,0,1,0)^{\mathrm{T}}$ |
| $-\frac{i}{2}(\varepsilon_1 + \varepsilon_2)$ | $(0,0,1,0)^{\mathrm{T}}$ | $(0,0,0,1)^{\mathrm{T}}$ |

The list of non-trivial solutions to this constraint is given in table 1.

For $m = \pm i\frac{\varepsilon_1 - \varepsilon_2}{2}$ two additional spinors of opposite chirality as compared to $\xi^{1,2}$ are preserved. This corresponds to twisting the $SU(2)_l \subset Spin(4)$ with the flavor symmetry so the full Lorentz group now is

$$[SU(2)_l \times SU(2)_F]_{\mathrm{diag}} \times [SU(2)_r \times SU(2)_R]_{\mathrm{diag}} \,. \tag{4.17}$$

This is the Marcus twist of $\mathcal{N} = 4$ SYM [18].

For $m = \pm i\frac{\varepsilon_1 + \varepsilon_2}{2}$ two additional spinors of the same chirality as $\xi^{1,2}$ are preserved. After the choice of this special mass, we have essentially identified $SU(2)_r \subset Spin(4)$ with the $SU(2)_R \times SU(2)_F$ so that the new Lorentz group is

$$SU(2)_l \times [SU(2)_r \times SU(2)_R \times SU(2)_F]_{\mathrm{diag}} \,. \tag{4.18}$$

This is the Vafa-Witten twist of $\mathcal{N} = 4$ SYM [19].

## 4.2 The four dimensional perspective and embedding in the $\mathcal{N} = 4$ supergravity

The analysis of the previous section nicely explains the symmetry enhancement for special masses but it does not give an inherently four-dimensional perspective. This is because the frame we used mixes the compact direction and the four-dimensions. We now rewrite the metric as

$$ds^2 = \left(\delta_{\mu\nu} - \frac{1}{1+v^2} v_\mu v_\nu\right) dx^\mu dx^\nu + \left(1 + v^2\right) \left(d\vartheta + \frac{v_\mu dx^\mu}{1+v^2}\right)^2 \,. \tag{4.19}$$

After dimensional reduction, the four-dimensional space has the metric

$$\delta_{\mu\nu} - \frac{v_\mu v_\nu}{1+v^2} \,. \tag{4.20}$$

The frame best suited to this split is

$$\widetilde{E}^a = dx^a - \frac{1}{v^2}\left(1 - \frac{1}{\sqrt{1+v^2}}\right) v^a v_\mu dx^\mu \,, \qquad \widetilde{E}^5 = \sqrt{1+v^2}\left(d\vartheta + \frac{v_a dx^a}{1+v^2}\right) \,. \tag{4.21}$$

This is related to our earlier frame by a local rotation implemented by the matrix with components

$$M^a{}_b = \delta_b^a - \frac{1}{v^2}\left(1 - \frac{1}{\sqrt{1+v^2}}\right) v^a v_b \,, \quad -M^a{}_5 = M^5{}_a = \frac{v_a}{\sqrt{1+v^2}} \,, \quad M^5{}_5 = \frac{1}{\sqrt{1+v^2}} \,. \tag{4.22}$$

The spinor transformation matrix can be computed from this and it can be shown to take the form

$$\mathcal{M} = \left(\frac{1-v^2}{1+v^2}\mathbf{1} - \frac{2}{1+v^2} v_\mu \Gamma^\mu \Gamma_5\right)^{\frac{1}{4}} \,. \tag{4.23}$$

Note that the Killing spinor equations do not change under this local frame rotation. The transformed spinors $\xi^i = \mathcal{M}\xi_0^i$ continue to satisfy eq. (4.9) but the spin-connection has changed. To facilitate the dimensional reduction we use the coordinate $\widetilde{\vartheta}$ such that $d\widetilde{\vartheta} = \sqrt{1+v^2}d\vartheta$. The spin-connection is

$$
\begin{aligned}
\Omega_{\mu,5b} &= e^\nu{}_b \partial_{[\nu}\left(\frac{v_{\mu]}}{\sqrt{1+v^2}}\right), \\
\Omega_{\widetilde{\vartheta},ab} &= -e^\mu{}_{[b}\Omega_{\mu,5a]}, \\
\Omega_{\mu,ab} &= \omega_{\mu,ab} - \frac{v_\mu}{\sqrt{1+v^2}}e^\nu{}_{[b}\Omega_{v5a]}.
\end{aligned}
\tag{4.24}
$$

The Killing spinors satisfy a constraint which follows from the $\widetilde{\vartheta}$-component of the Killing spinor equation

$$
\Omega_{a,5b}\Gamma^{ab}\xi^i = \frac{2i}{\sqrt{1+v^2}}(\varepsilon_1+\varepsilon_2)(\tau_3)^i{}_j\xi^j.
\tag{4.25}
$$

Using this in the $\mu$-component of eq. (4.9) we obtain the following four-dimensional equation satisfied by the Killing spinors

$$
\nabla_\mu\xi^i - \frac{i(\varepsilon_1+\varepsilon_2)}{2(1+v^2)}v_\mu(\tau_3)^i{}_j\xi^j + \frac{1}{8}\Omega_{a5b}\Gamma^{ab}\Gamma_\mu\Gamma^5\xi^i = \frac{i(\varepsilon_1+\varepsilon_2)}{4\sqrt{1+v^2}}\Gamma_\mu(\tau_3)^i{}_j\Gamma^5\xi^i.
\tag{4.26}
$$

Comparing the above with the generalized Killing spinor equation that arises from setting the variation of the gravitino to zero in the $\mathcal{N}=2$ conformal supergravity, eqs. (A.4) and (A.5), we can read off the $\mathcal{N}=2$ supergravity background fields. These are given by

$$
V_\mu = \frac{i}{2}\frac{\varepsilon_1+\varepsilon_2}{1+v^2}v_\mu\tau_3, \quad B_{\mu\nu} = -i\Omega_{\mu,5\nu}, \quad \overline{\eta}^i = \frac{\varepsilon_1+\varepsilon_2}{2\sqrt{1+v^2}}(\tau_3)^i{}_j\Gamma^5\xi^i + \frac{1}{8}\Omega_{\mu,5\nu}\Gamma^{\mu\nu}\Gamma^5\xi^i. \tag{4.27}
$$

Let us now embed these as constraints on a supersymmetric background of $\mathcal{N}=4$ conformal supergravity. The bosonic fields in the $\mathcal{N}=4$ Weyl multiplet are the vierbein $e^a{}_\mu$, a two-form $T^{IJ}_{ab}$, an $SU(4)_R$ gauge field $V_\mu{}^I{}_J$, a one form $b_\mu$, and scalars $E_{IJ}, D^{IJ}{}_{KL}, C$. The indices $I, J, ..$ run from 1 to 4. Following [39, 40] this provides a supersymmetric background if we require the fermions of the multiplet and their SUSY transformations to vanish. The relevant equations for us are

$$
0 = \delta\Lambda_I = E_{IJ}\epsilon^J - \frac{1}{2}\epsilon_{IJKL}\gamma \cdot T^{KL}\epsilon^J,
\tag{4.28}
$$

$$
0 = \delta\psi_\mu^I = 2\nabla_\mu\epsilon^I - 2V_{\mu}^I{}_J\epsilon^J - \frac{i}{2}\gamma \cdot T^{IJ}\gamma_\mu\epsilon_J + i\gamma_\mu\eta^I.
\tag{4.29}
$$

Note that we are using the chiral $SU(4)$ conventions [39], i.e. $\epsilon^I, \eta_I$ are chiral and $\epsilon_I, \eta^I$ are anti-chiral. Also there is a second set of charge conjugate equations. It is now straightforward to interpret eq. (4.25) and eq. (4.26) as arising from the above supersymmetric variations. For example when only $\mathcal{N}=2$ supersymmetry is preserved, i.e., $\epsilon^3 = \epsilon^4 = 0$, various parameters are related as follows.

$$
\begin{aligned}
&\epsilon^1 \pm \epsilon^2 = \frac{1+\Gamma^5}{2}\xi^{1,2}, \quad \epsilon_1 \pm \epsilon_2 = \frac{1-\Gamma^5}{2}\xi^{2,1}, \quad \eta^1 \pm \eta^2 = \frac{\varepsilon_1+\varepsilon_2}{2\sqrt{1+v^2}}\frac{1+\Gamma^5}{2}\xi^{1,2} \\
&T^{12} = T^{34} = -\frac{1}{2}B, \quad E_{11} = -E_{22} = -\frac{\varepsilon_1+\varepsilon_2}{\sqrt{1+v^2}}, \quad V_\mu = i\frac{\varepsilon_1+\varepsilon_2}{8(1+v^2)}v_\mu\begin{pmatrix}\tau_1 & 0 \\ 0 & 0\end{pmatrix}.
\end{aligned}
\tag{4.30}
$$

Similarly the enhanced supersymmetric background can be embedded in the $\mathcal{N}=4$ supergravity when $\epsilon^3$ and $\epsilon^4$ are linear combinations of the Killing spinors $\xi^{3,4}$.

### 4.3 Linearized $\Omega$-background

Let us conclude this section by relating the $\Omega$-background we obtained to the one discussed in [27, 41]. The origin is a fixed point of $v$ around which it generates a $T^2$ action. To leading order in the distance from the origin

$$V_\mu = 0\,, \qquad B_{\mu\nu} = \frac{\mathrm{i}}{2}\left(\partial_\mu v_\nu - \partial_\nu v_\mu\right)\,. \tag{4.31}$$

The linearized Killing spinor equation is

$$\partial_\mu \xi^i - \frac{1}{8}\partial_\rho v_\nu \left[\Gamma^{\rho\nu}, \Gamma_\mu\right]\Gamma^5 \xi_0^i = 0\,, \tag{4.32}$$

which is satisfied by

$$\xi^i = \left(\mathbf{1} - \frac{1}{2}v^\mu \Gamma_\mu \Gamma^5\right)\xi_0^i\,. \tag{4.33}$$

These are precisely the equations that are referred to as the $\Omega$-background by [27, 41]. Here we see that these arise as a linear approximation to the $\Omega$-background which is obtained by dimensional reduction with a twist.

## 5 Applications

We now discuss various applications of our results. Some simplifications, observed in the literature, of the free energy of four-dimensional supersymmetric theories can be explained by our results. Using the AGT correspondence our results shed light on the torus vacuum and the one-point function of certain *local* operators in Liouville/Toda CFTs. Finally, our results imply an infinite number of constraints between correlators of operators at fixed radial distance from the origin but integrated over the other directions. This generalizes relations obtained earlier between integrated correlators of $\mathcal{N} = 4$ SYM.

### 5.1 Squashing independence of the free energy

It was pointed out in [7] that the free energy of the mass deformed $\mathcal{N} = 4$ SYM on the ellipsoid with $U(1) \times U(1)$-isometry is independent of the squashing parameter $b = \sqrt{\frac{\ell}{\widetilde{\ell}}}$ if the mass of the adjoint hypermultiplet is

$$m_* = \pm \frac{\mathrm{i}}{\sqrt{\ell\widetilde{\ell}}}\frac{b - b^{-1}}{2} = \pm \frac{\mathrm{i}}{2}\left(\ell^{-1} - \widetilde{\ell}^{-1}\right)\,. \tag{5.1}$$

We now show that this value of the mass precisely corresponds to the special values which lead to symmetry enhancement at the two poles of the squashed sphere.

The north pole of the squashed sphere is a *plus* fixed point characterized by equivariant parameters $\frac{1}{\ell}$ and $\frac{1}{\widetilde{\ell}}$, so the Marcus point corresponds to mass $\pm\frac{\mathrm{i}}{2}\left(\ell^{-1} - \widetilde{\ell}^{-1}\right) = m_*$ at the north pole. The south pole of the sphere is a *minus* fixed point with equivariant parameters [12] $\frac{1}{\ell}$ and $-\frac{1}{\widetilde{\ell}}$ so the corresponding Marcus point occurs at $\pm\frac{\mathrm{i}}{2}\left(\ell^{-1} + \left(-\widetilde{\ell}\right)^{-1}\right) = m_*$. We see that Marcus point at both fixed points happens to agree with the constant choice of mass at which the simplification of the partition function was observed.

Similarly, the constant mass $\pm\frac{\mathrm{i}}{2}\left(\ell^{-1} + \widetilde{\ell}^{-1}\right)$ corresponds to the Vafa-Witten point at both fixed points. This has not been emphasized earlier in the literature. For this special value of the mass the partition function is the same as in the case of the round sphere with mass $\pm\mathrm{i}$ [8].

This local symmetry enhancement near the fixed points of $v$ is enough to explain the simplifications at hand. One might still wonder about a global enhancement on the squashed sphere. However this makes matters much more complicated. First the position dependence of the two-form field implies through (4.28) that a constant mass will no longer do the trick. This position dependence of the mass requires additional non-trivial components of the background vector field as we have shown in section 3.1. A similar requirement comes from integrating the full covariant version of the gravitino equation (4.29).

## 5.2 AGT correspondence

In class $\mathcal{S}$ theories [42, 43], the complex curve associated to the $\mathcal{N} = 2^*$ theory is the once-punctured torus. The AGT-correspondence [20] relates the observables of $\mathcal{N} = 2^*$ theory on the squashed sphere to those of Liouville/Toda theory on torus $T^2$ with complex structure identified with the gauge coupling $\tau$. For a review on class $\mathcal{S}$ and the AGT correspondence we refer to [44].

For the gauge group $SU(2)$, the AGT correspondence states that

$$\mathcal{Z}_{\mathcal{N}=2^*}(\tau, \mu, b) = \langle \widehat{V}_{\frac{Q}{2}+i\mu} \rangle_{T_\tau^2}, \tag{5.2}$$

where $Q = b + b^{-1}$. The left-hand side is the partition function of $\mathcal{N} = 2^*$ theory with dimensionless mass parameter $\mu = \sqrt{\ell\tilde{\ell}}m$ on the squashed sphere, while the right hand side is the one-point function of the vertex operator of dimension $h = \bar{h} = \frac{Q^2}{4} + \mu^2$ in Liouville theory on the torus of modular parameter $\tau$. To derive this it is sufficient to compare the partition function with the structure constants from the DOZZ formula

$$C(\alpha_1, Q-\alpha_1, \alpha) = \left[\pi\mu\gamma(b^2)b^{2-2b^2}\right]^{-\alpha/b} \frac{\Upsilon(b)\Upsilon(2\alpha_1)\Upsilon(2(Q-\alpha_1))\Upsilon(2\alpha)}{\Upsilon(\alpha)\Upsilon(Q-2\alpha_1+\alpha)\Upsilon(2\alpha_1-Q+\alpha)\Upsilon(Q-\alpha)}, \tag{5.3}$$

with $\Upsilon(x) = \prod_{m,n\geq 0}((mb+\frac{n}{b})+x)((m+1)b+\frac{n+1}{b}-x)$. Matching $2\alpha_1-Q$ with the Coulomb branch parameter $-i\sqrt{\frac{\ell\tilde{\ell}}{r^2}}\sigma$ and $\alpha$ with $\frac{Q}{2}+i\mu$ we see that the vertex operator takes the form $\widehat{V}_\alpha = \frac{\Upsilon(\alpha)}{\Upsilon(2\alpha)}\left[\pi\mu\gamma(b^2)b^{2-2b^2}\right]^{\alpha/b}e^{2\alpha\phi}$.

We are interested in the case where $\mu$ is purely imaginary, which corresponds to *local* operators but non-normalizable states [45]. For $\mu = \pm i\frac{b-b^{-1}}{2}$ we get that $h = \bar{h} = 1$, i.e. the one-point functions of the exactly marginal operators $\widehat{V}_b, \widehat{V}_{b^{-1}}$ are independent of the parameter $b$. On the other hand, for $\mu = \pm i\frac{b+b^{-1}}{2}$ we have $h = \bar{h} = 0$. The vertex operators $\widehat{V}_0, \widehat{V}_Q$ are proportional to the identity operator in the CFT and we conclude that the torus partition function of the Liouville theory does not depend on $b$ either.

The generalization to generic gauge groups involves Toda theory [46]. Following this idea, we should compare the squashed sphere partition function with the Toda structure constant [47],

$$C(\alpha_1, 2Q-\alpha_1, k\omega_{N-1}) \tag{5.4}$$

$$= \left[\pi\mu\gamma(b^2)b^{2-2b^2}\right]^{-\frac{k(N-1)}{2b}} \frac{(\Upsilon(b))^{N-1}\Upsilon(k)\prod_{e\in\Delta_+}\Upsilon(\langle Q-\alpha_1, e\rangle)\Upsilon(\langle\alpha_1-Q, e\rangle)}{\prod_{i,j=1}^N \Upsilon\left(\frac{k}{N}+\langle\alpha_1-Q, h_i\rangle+\langle(Q-\alpha_1, h_j\rangle\right)} \tag{5.5}$$

$$= \left[\pi\mu\gamma(b^2)b^{2-2b^2}\right]^{-\frac{k(N-1)}{2b}} \frac{(\Upsilon(b))^{N-1}\Upsilon(k)}{\Upsilon\left(\frac{k}{N}\right)^N} \prod_{e\in\Delta_+} \frac{\Upsilon(\langle Q-\alpha_1, e\rangle)\Upsilon(\langle\alpha_1-Q, e\rangle)}{\Upsilon\left(\frac{k}{N}+\langle\alpha_1-Q, e\rangle\right)\Upsilon\left(\frac{k}{N}-\langle\alpha_1-Q, e\rangle\right)}. \tag{5.6}$$

Here $\langle\bullet,\bullet\rangle$ is the inner product on the root space, $h_i$ are the weights of fundamental representation of $SU(N)$, $k$ is a constant, $Q = (b+b^{-1})\rho$ for $\rho$ the Weyl vector, $\alpha$ parametrizes

a vertex operator $\widehat{V}_\alpha = e^{\langle \alpha, \phi \rangle}$, and $\omega_{N-1}$ is last element in the basis dual to the simple roots. Identifying $\alpha_1 - Q = i\sqrt{\frac{\ell\bar{\ell}}{r^2}}a$ and $\frac{k}{N} = \frac{b+b^{-1}}{2} + i\mu$, the correspondence states

$$Z_{\mathcal{N}=2^*}(\tau, m, b) = \langle \widehat{V}_{k\omega_{N-1}} \rangle_{T^2_\tau}, \tag{5.7}$$

for the vertex operator $\widehat{V}_{k\omega_{N-1}} = \frac{\Upsilon(k/N)}{\Upsilon(k)} \left[ \pi\mu\gamma(b^2)b^{2-2b^2} \right]^{k(N-1)/(2b)} e^{\langle k\omega_{N-1}, \phi \rangle}$. The dimension of this vertex operator is

$$\Delta(k\omega_{N-1}) = \frac{N-1}{2}k\left( \left( b + b^{-1} \right) - \frac{k}{N} \right). \tag{5.8}$$

With this at hand we can once again translate the squashing independence of the partition function into CFT. For the imaginary masses $\mu$, $k$ is real and thus the operators $\widehat{V}_{k\omega_{N-1}}$ do not correspond to normalizable states in the spectrum of Toda theory [48, 49]. Based on what we know from Liouville theory we still expect them to be sensible *local* operators. The $\mathcal{N} = 2^*$ at the Marcus point implies that the one-point functions of the dimension $\frac{1}{2}N(N-1)$ operators $\widehat{V}_{bN\omega_{N-1}}$ and $\widehat{V}_{b^{-1}N\omega_{N-1}}$ are independent of $b$. The Vafa-Witten point corresponds to the operators $\widehat{V}_0, \widehat{V}_{QN\omega_{N-1}}$. These two have dimension zero and are thus proportional to the identity operator in the Toda theory. We find that their one-point function, and thus the Toda torus partition function, does not depend on $b$.

## 5.3 Constraints on correlators

An application of the position-dependent flavor deformation we introduced in section 3 is to obtain relations between integrated correlators in $\mathcal{N} = 2$ SCFTs with flavor symmetries and marginal couplings. This implies improvement on the results of [7, 21–24, 50–53] for integrated correlators of $\mathcal{N} = 4$ SYM[8]. These correlators are obtained by taking derivatives of the localized mass-deformed theory with respect to couplings and masses. The appearance of a position-dependent mass allows one to take functional derivatives with respect to it and refine the earlier results. A detailed study of the additional information that can in this way be extracted is left for future work. Here we present the basic framework and an example of how our results can be used to refine earlier constraints on integrated correlators.

The position-dependent flavor deformation is introduced by coupling the flavor-current multiplet with bosonic components $\left( j_{a,\mu}, \Sigma_a, \overline{\Sigma}_a, \mathcal{B}_a^{ij} \right)$ to the background vector multiplet $\left( A_a^\mu, m_a, \overline{m}_a, D_{a,ij} \right)$ where $a = 1, 2, \cdots, r_F$. The precise coupling works out to be

$$\mathcal{L}_{\text{def}} = \sum_{a=1}^{r_F} \frac{1}{2} A_a^\mu j_{a,\mu} + \frac{1}{2} D_{a,ij} \mathcal{B}_a^{ij} + \frac{1}{2} m_a \Sigma_a + \frac{1}{2} \overline{m}_a \overline{\Sigma}_a + \left( \frac{1}{2} A_a^2 - 2 m_a \overline{m}_a \right) L_a. \tag{5.9}$$

The last, non-minimal term is the usual mass-term for hypermultiplet scalars.

In general, the masses can depend on the three directions transverse to the Killing vector $v$. To simplify the following discussion we will however restrict to the round sphere $\mathbb{S}^4$ with the metric

$$ds^2 = r^2 \left( d\rho^2 + \sin^2\rho\, d\theta^2 + \sin^2\rho \sin^2\theta\, d\phi^2 + \sin^2\rho \cos^2\theta\, d\chi^2 \right) \tag{5.10}$$

and masses that only depend on the latitudinal coordinate $\rho$. Moreover let us assume $\widetilde{s}m_a + s\overline{m}_a = M_a$ is constant such that the background gauge field vanishes. Using $s = \sin^2\frac{\rho}{2}$, $\widetilde{s} = \cos^2\frac{\rho}{2}$ we then get

$$m_a = M_a + is\varphi_a(\rho), \tag{5.11}$$

$$\overline{m}_a = M_a - i\widetilde{s}\varphi_a(\rho). \tag{5.12}$$

---

[8]We thank J. A. Minahan and R. Panerai for suggesting the application of position-dependent masses to obtain constraints on correlators.

The partition function, which only depends on the values of $m_a$ and $\overline{m}_a$ at north and the south poles, is independent of $\varphi_a(\rho)$.

Before progressing to the partition function, we first discuss the functional derivatives of the deformation action. The first such derivative has the form

$$\frac{\delta S_{\text{def}}}{\delta \varphi_a(\rho)} = \int \sqrt{g} \mathrm{d}^4 x \left[ \frac{\mathrm{i}s}{2}\delta(\rho(x)-\rho)\Sigma_a(x) + \frac{\widetilde{\mathrm{i}s}}{2}\delta(\rho(x)-\rho)\overline{\Sigma}_a(x) + \sum_{b=1}^{r_F} \frac{1}{2}\frac{\delta D_{b,ij}(x)}{\delta \varphi_a(\rho)}\mathcal{B}_b^{ij}(x) \right.$$
$$\left. -2(\mathrm{i}s\overline{m}_a(x)-\widetilde{\mathrm{i}s}m_a(x))\delta(\rho(x)-\rho)L_a(x) \right]. \quad (5.13)$$

Due to the derivatives on the mass in (3.4) the functional derivative of $D$ is non-trivial. We do not evaluate it explicitly here, but we note that by integration by parts we get

$$\sum_{b=1}^{r_F} \int \sqrt{g} \mathrm{d}^4 x \frac{\delta D_{b,ij}(x)}{\delta \varphi_a(\rho)}\mathcal{B}_b^{ij}(x) = \int \sqrt{g}\mathrm{d}^4 x \left( \widetilde{D}_{ij}(x)\mathcal{B}_a^{ij}(x)\delta(\rho(x)-\rho) + \widetilde{D}_{ij}^\mu(x)\partial_\mu \mathcal{B}_a^{ij}(x)\delta(\rho(x)-\rho) \right),$$
$$(5.14)$$

for some $\widetilde{D}_a, \widetilde{D}_a^\mu$. This introduces the first descendant of $\mathcal{B}$. The second functional derivative of the action is

$$\frac{\delta^2 S_{\text{def}}}{\delta \varphi_a(\rho)\delta \varphi_b(\rho')} = -2\delta(\rho-\rho')\delta^{ab}s(\rho)\widetilde{s}(\rho)\int \sqrt{g}\mathrm{d}^3 x L_b(x,\rho). \quad (5.15)$$

With all of this at hand we are ready to look at the correlators we can get from the partition function. The simplest relation that we obtain involves four derivatives of the partition function with respect to $\varphi(\rho)$ and marginal parameters[9]

$$\partial_{\tau_i}\partial_{\overline{\tau}_j}\frac{\delta^2 Z}{\delta \varphi_a(\rho)\delta \varphi_b(\rho')}\bigg|_{\varphi=0} = 0. \quad (5.16)$$

We emphasize that this relation holds for *any* $\mathcal{N}=2$ SCFT with marginal parameters and flavor symmetry. As an example we demonstrate how this works for $\mathcal{N}=4$ SYM. The theory has only one marginal parameters and $r_F = 1$. Using eqs. (2.6) and (5.13) to (5.15) we can express the left hand side of eq. (5.16) in terms of various correlators of $\mathcal{N}=4$ SYM operators. Using the selection rule that the correlators with odd number of $\Sigma$ or $\overline{\Sigma}$ vanish the resulting constraint can be written as

$$0 = -32\pi^2 r^4 \delta(\rho-\rho')s\widetilde{s}(\rho)\int \sqrt{g}\mathrm{d}x^3 \langle \text{Tr}(X^2)(0)\text{Tr}(\overline{X}^2)(\pi)L(x,\rho)\rangle$$

$$-\pi^2 r^4 s(\rho)\widetilde{s}(\rho')\iint \sqrt{g}\sqrt{g'}\mathrm{d}x^3 \mathrm{d}x'^3 \left\langle \text{Tr}(X^2)(0)\text{Tr}(\overline{X}^2)(\pi)\Sigma(x,\rho)\overline{\Sigma}(x',\rho') \right\rangle + \rho \leftrightarrow \rho'$$

$$+4\pi^2 r^4 \iint \sqrt{g}\sqrt{g'}\mathrm{d}x^3 \mathrm{d}x'^3 \left( \widetilde{D}_{ij}(x,\rho) + \widetilde{D}_{ij}^\mu(x,\rho)\frac{\partial}{\partial x^\mu} \right)\left( \widetilde{D}_{ij}(x',\rho') + \widetilde{D}_{ij}^\mu(x',\rho')\frac{\partial}{\partial x'^\mu} \right)$$

$$\times \left\langle \text{Tr}(X^2)(0)\text{Tr}(\overline{X}^2)(\pi)\mathcal{B}^{ij}(x,\rho)\mathcal{B}^{ij}(x',\rho') \right\rangle. \quad (5.17)$$

The superconformal Ward identities for the $\mathcal{N}=4$ stress tensor multiplet allow one to write all of the above 4-point functions in terms of differential operators acting on a single function $\mathcal{T}(U,V)$ [21,54,55], of the two conformal cross ratios $U, V$. The latter cross ratio only depends on the coordinates $\rho, \rho'$. As a consequence it is not integrated over in contrast to the case of constant mass. Thus, the above relation yields a differential equation for $\mathcal{T}$. We emphasize that (5.17) is a general consequence of $\mathcal{N}=2$ supersymmetry and repackages the complicated constraints from Ward Identities as linear relations between integrated correlation functions.

---

[9]The relations with less than three derivatives have counter-term ambiguities and the relation with three derivatives are trivially satisfied.

## Acknowledgements

We thank Guido Festuccia and Joseph Minahan for comments on an earlier draft and valuable suggestions. Results in sec. 5.3 were developed in collaboration with Joseph Minahan and Rodolfo Panerai. We are indebted to Yifan Wang for numerous critical comments and correspondence which led to the improvement of the content. This research is supported in part by Vetenskapsrådet under grants #2016-03503, #2018-05572 and #2020-03339, and by the Knut and Alice Wallenberg Foundation under grant Dnr KAW 2015.0083.

## A $\mathcal{N} = 2$ on four manifolds

In this Appendix we give all the important expressions for the $\mathcal{N} = 2$ backgrounds introduced in [10].

At the basis of the construction is a covering of the manifold with open patches such that each patch contains at most one fixed point of $v$. In patches where $s \neq 0$ we can define spinors

$$\zeta^i_\alpha = \frac{\sqrt{s}}{2}\delta^i_\alpha, \qquad \overline{\chi}_i = \frac{1}{s}v^\mu\overline{\sigma}_\mu\zeta_i. \tag{A.1}$$

Transitioning into other patches with $s \neq 0$, we should undo the required $SU(2)_l$ transformation on $\zeta$ by an $SU(2)_R$ transformation so that $\zeta$ keeps the same expression. $\overline{\chi}$ will then also transform correctly. To transform into a patch with a zero of $s$ one should do besides the $SU(2)_l$ transformation the $SU(2)_R$ transformation given by

$$U_i{}^j = i\frac{v^\mu}{\|v\|}\sigma_{\mu i}{}^j, \tag{A.2}$$

such that in this patch we have

$$\zeta_i = -\frac{1}{\tilde{s}}v^\mu\sigma_\mu\overline{\chi}_i, \quad \overline{\chi}^{\dot\alpha} = -i\frac{\sqrt{\tilde{s}}}{2}\delta^i_\alpha, \tag{A.3}$$

This gives a globally well defined set of spinors on the four-manifold.

The Killing spinor equations read

$$(D_\mu - iG_\mu)\zeta_i - \frac{i}{2}W^+_{\mu\rho}\sigma^\rho\overline{\chi}_i - \frac{i}{2}\sigma_\mu\overline{\eta}_i = 0, \tag{A.4}$$

$$(D_\mu + iG_\mu)\overline{\chi}^i + \frac{i}{2}W^-_{\mu\rho}\overline{\sigma}^\rho\zeta^i - \frac{i}{2}\overline{\sigma}_\mu\eta^i = 0, \tag{A.5}$$

and the auxiliary equations are

$$(N - \frac{1}{6}R)\overline{\chi}^i = 4i\partial_\mu G_\nu\overline{\sigma}^{\mu\nu}\overline{\chi}^i + i(\nabla^\mu + 2iG^\mu)W^-_{\mu\nu}\overline{\sigma}^\nu\zeta^i + i\overline{\sigma}^\mu(D_\mu + iG_\mu)\eta^i, \tag{A.6}$$

$$(N - \frac{1}{6}R)\zeta_i = -4i\partial_\mu G_\nu\sigma^{\mu\nu}\zeta_i - i(\nabla^\mu - 2iG^\mu)W^+_{\mu\nu}\sigma^\nu\overline{\chi}_i + i\sigma^\mu(D_\mu - iG_\mu)\overline{\eta}_i. \tag{A.7}$$

The auxiliary Killing spinors take the form

$$\eta_i = (\mathcal{F}^+ - W^+)\zeta_i - G_\mu\sigma^\mu\overline{\chi}_i - S_{ij}\zeta^j, \tag{A.8}$$

$$\overline{\eta}^i = -(\mathcal{F}^- - W^-)\overline{\chi}^i + 2G_\mu\overline{\sigma}^\mu\zeta^i - S^{ij}\overline{\chi}_j, \tag{A.9}$$

where we use the notation $W^+ = \frac{1}{2}W_{\mu\nu}\sigma^{\mu\nu}$ and $W^- = \frac{1}{2}W_{\mu\nu}\overline{\sigma}^{\mu\nu}$ and similar for $\mathcal{F}$.

With the notations

$$v_\mu^{(ij)} = \zeta^i \sigma_\mu \overline{\chi}^j + \zeta^j \sigma_\mu \overline{\chi}^i, \tag{A.10}$$

$$\Theta_{\mu\nu}^{(ij)} = \zeta^j \sigma_{\mu\nu} \zeta^i, \tag{A.11}$$

$$\tilde{\Theta}_{\mu\nu}^{(ij)} = \overline{\chi}^i \overline{\sigma}_{\mu\nu} \overline{\chi}^j, \tag{A.12}$$

the background fields take the form

$$W_{\mu\nu} = \frac{i}{s+\tilde{s}}(\partial_\mu v_\nu - \partial_\nu v_\mu) - \frac{2i}{(s+\tilde{s})^2}\epsilon_{\mu\nu\rho}{}^{\lambda}v^\rho \partial_\lambda(s-\tilde{s}) - \frac{4}{s+\tilde{s}}\epsilon_{\mu\nu\rho}{}^{\lambda}v^\rho G_\lambda \tag{A.13}$$

$$+ \frac{s-\tilde{s}}{(s+\tilde{s})^2}\epsilon_{\mu\nu\rho}{}^{\lambda}v^\rho b_\lambda + \frac{1}{s+\tilde{s}}(v_\mu b_\nu - v_\nu b_\mu), \tag{A.14}$$

$$(V_\mu)_{ij} = \frac{4}{s+\tilde{s}}(\zeta_{(i}\nabla_\mu \zeta_{j)} + \overline{\chi}_{(i}\nabla_\mu \overline{\chi}_{j)}) + \frac{4}{s+\tilde{s}}\left(2iG_\nu - \frac{\partial_\nu(s-\tilde{s})}{(s+\tilde{s})}\right)(\Theta_{ij} - \tilde{\Theta}_{ij})^\nu{}_\mu \tag{A.15}$$

$$+ \frac{4i}{(s+\tilde{s})^2}b_\nu(\tilde{s}\Theta_{ij} + s\tilde{\Theta}_{ij})^\nu{}_\mu, \tag{A.16}$$

$$\mathcal{F}_{\mu\nu} = i\partial_\mu\left(\frac{s+\tilde{s}-K}{s\tilde{s}}v_\nu\right) - i\partial_\nu\left(\frac{s+\tilde{s}-K}{s\tilde{s}}v_\mu\right), \tag{A.17}$$

$$S_{ij} = \frac{8i}{(s+\tilde{s})^2}(\Theta_{ij} + \tilde{\Theta}_{ij})^{\mu\nu}\partial_\mu v_\nu - 2i\frac{s+\tilde{s}-K}{(s\tilde{s})^2}(\tilde{s}\Theta_{ij} + s\tilde{\Theta}_{ij})^{\mu\nu}\partial_\mu v_\nu$$
$$+ \frac{2}{s+\tilde{s}}\left(4G_\mu - \frac{s-\tilde{s}}{s+\tilde{s}}b_\mu - \frac{i}{2}\frac{s-\tilde{s}}{s\tilde{s}}\partial_\mu(s+\tilde{s})\right)v_{ij}^\mu. \tag{A.18}$$

For the scalar $R/6-N$ we refer to the original paper as it is not necessary for our computations.

A couple of necessary conventions are

$$\sigma_{\alpha\dot{\alpha}}^\mu = (\vec{\tau}, -i),$$
$$\overline{\sigma}_{\alpha\dot{\alpha}}^\mu = (-\vec{\tau}, -i),$$
$$\sigma^{\mu\nu} = \frac{1}{4}(\sigma_\mu\overline{\sigma}_\nu - \sigma_\nu\overline{\sigma}_\mu),$$
$$\overline{\sigma}^{\mu\nu} = \frac{1}{4}(\overline{\sigma}_\mu\sigma_\nu - \overline{\sigma}_\nu\sigma_\mu),$$

with the Pauli-matrices $\vec{\tau} = (\tau^1, \tau^2, \tau^3)$. Note also that $\epsilon_{1234} = \epsilon^{12} = -\epsilon_{12} = 1$, where the former is used to define duality of tensor while the latter two are used to raise and lower $SU(2)$ indices.

## A.1  Cohomological formulation

The cohomological fields are defined as:

$$\eta = \zeta_i \lambda^i + \overline{\chi}^i \overline{\lambda}_i, \tag{A.19}$$

$$\varphi = -i(X - \overline{X}), \tag{A.20}$$

$$\Psi_\mu = \zeta_i \sigma_\mu \overline{\lambda}^i + \overline{\chi}^i \overline{\sigma}_\mu \lambda_i, \tag{A.21}$$

$$\phi = \tilde{s}X + s\overline{X}, \tag{A.22}$$

$$\chi_{\mu\nu} = 2\frac{s+\tilde{s}}{s^2+\tilde{s}^2}\left(\overline{\chi}^i\overline{\sigma}_{\mu\nu}\overline{\lambda}_i - \zeta_i\sigma_{\mu\nu}\lambda^i + \frac{1}{s+\tilde{s}}(v_\mu\Psi_\nu - v_\nu\Psi_\mu)\right), \tag{A.23}$$

$$H_{\mu\nu} = \left(P_{\omega_c}^+\right)_{\mu\nu}^{\rho\lambda}\left[\widehat{\Theta}_{\rho\lambda}^{ij}D_{ij} - F_{\rho\lambda} + i\frac{X+\overline{X}}{s+\tilde{s}}(\partial_\rho v_\lambda - \partial_\lambda v_\rho)\right.$$
$$\left.- \frac{2i}{s+\tilde{s}}\epsilon_{\rho\lambda\gamma}{}^{\delta}v^\gamma\left(\left(D_\delta - 2iG_\delta - i\frac{\tilde{s}}{s+\tilde{s}}b_\delta\right)X - \left(D_\delta + 2iG_\delta - i\frac{s}{s+\tilde{s}}b_\delta\right)\overline{X}\right)\right]. \tag{A.24}$$

This last definition uses the notations

$$\cos(\omega_c) = \frac{s - \tilde{s}}{s + \tilde{s}}, \tag{A.25}$$

$$\kappa = g(v, \bullet), \tag{A.26}$$

$$P^+_{\omega_c} = \frac{1}{1 + \cos^2 \omega_c} \left( 1 + \cos \omega_c \star - \sin^2 \omega_c \frac{\kappa \wedge \iota_v}{\iota_v \kappa} \right), \tag{A.27}$$

$$\widehat{\Theta}^{ij}_{\mu\nu} = \frac{4}{1 + \cos^2 \omega_c} \left( \frac{\cos^2(\omega_c/2)}{s} \Theta^{ij}_{\mu\nu} + \frac{\sin^2(\omega_c/2)}{\tilde{s}} \widetilde{\Theta}^{ij}_{\mu\nu} \right). \tag{A.28}$$

After this field redefinition the SUSY transformations of the vector multiplet take the form:

$$\delta A = i\Psi, \tag{A.29}$$

$$\delta \Psi = \iota_v \Psi + i d_A \phi, \tag{A.30}$$

$$\delta \phi = \iota_v \Psi, \tag{A.31}$$

$$\delta \varphi = i\eta, \tag{A.32}$$

$$\delta \eta = L^A_v \varphi - [\phi, \varphi], \tag{A.33}$$

$$\delta \chi = H, \tag{A.34}$$

$$\delta H = i L^A_v \chi - i[\phi, \chi]. \tag{A.35}$$

# B  Including fluxes

The localization results we are using and by extension our results also hold on manifolds that support non-trivial fluxes such as $\mathbb{RP}^4$ [56]. In this appendix we give the corresponding expressions for the one-loop determinants and the partition functions. We refer to [11] for a detailed discussion.

Inclusion of non-trivial fluxes in the localized partition function shifts the Coulomb branch parameter in the classical, Nekrasov and one-loop contributions to the partition function. For the classical action this precisely gives the corresponding contribution $N(\{k_i\})$ to the instanton number

$$\log Z_{\text{classical}} = -\frac{16\pi^2}{g^2_{\text{YM}}} \text{Tr}(\sigma^2) s_- - 2\pi i \tau N(\{k_i\}), \tag{B.1}$$

where the index $i$ runs over the number of fixed points of $v$. For the one-loop and Nekrasov partition functions the shift is $\sigma \to \sigma + k_i(\varepsilon_i, \varepsilon'_i)$ with the function $k_i$ valued in the Cartan of $\mathfrak{g}$ encoding the flux contribution at the $i$-th fixed point. For the vector multiplet this leads to a modification of the general expressions for the two types of fixed points, which now become

$$Z^{\text{vec}}_{\varepsilon_x, \varepsilon'_x}(\sigma) = \prod_{\rho \in \mathbf{adj}} \prod_{n_1, n_2 \geq 0} \left( i\rho(\sigma) + i\rho(k_x(\varepsilon_x, \varepsilon'_x)) + (n_1 + 1)\varepsilon_x + (n_2 + 1)\varepsilon'_x \right)^{\frac{1}{2}}$$

$$\times \prod_{\substack{m_1, m_2 \geq 0 \\ (m_1, m_2) \neq (0,0)}} \left( i\rho(\sigma) + i\rho(k_x(\varepsilon_x, \varepsilon'_x)) + m_1 \varepsilon_x + m_2 \varepsilon'_x \right)^{\frac{1}{2}},$$

$$Z^{\text{vec}}_{\varepsilon_y, \varepsilon'_y}(\sigma) = \prod_{\rho \in \mathbf{adj}} \prod_{n_1, n_2 \geq 0} \left( i\rho(\sigma) + i\rho(k_y(\varepsilon_y, \varepsilon'_y)) - (n_1 + 1)\varepsilon_y + (n_2 + 1)\varepsilon'_y \right)^{\frac{1}{2}}$$

$$\times \prod_{\substack{m_1, m_2 \geq 0 \\ (m_1, m_2) \neq (0,0)}} \left( i\rho(\sigma) + i\rho(k_y(\varepsilon_y, \varepsilon'_y)) - m_1 \varepsilon_y + m_2 \varepsilon'_y \right)^{\frac{1}{2}}. \tag{B.2}$$

Similarly the hypermultiplet expressions become

$$
\begin{aligned}
Z^{\text{hyp}}_{\varepsilon_x,\varepsilon'_x}(\sigma, m_x) = \prod_{\rho\in\mathbf{R}}\prod_{n_1,n_2\geq 0} &\left(i\rho(\sigma)+i\rho(k_x(\varepsilon_x,\varepsilon'_x))+im_x+(n_1+\tfrac{1}{2})\varepsilon_x+(n_2+\tfrac{1}{2})\varepsilon'_x\right)^{-\frac{1}{2}}\\
&\times\left(-i\rho(\sigma)-i\rho(k_x(\varepsilon_x,\varepsilon'_x))-im_x+\left(n_1+\tfrac{1}{2}\right)\varepsilon_x+\left(n_2+\tfrac{1}{2}\right)\varepsilon'_x\right)^{-\frac{1}{2}},\\
Z^{\text{hyp}}_{\varepsilon_y,\varepsilon'_y}(\sigma, m_y) = \prod_{\rho\in\mathbf{R}}\prod_{n_1,n_2\geq 0} &\left(i\rho(\sigma)+i\rho(k_y(\varepsilon_y,\varepsilon'_y))+im_y-\left(n_1+\tfrac{1}{2}\right)\varepsilon_y+(n_2+\tfrac{1}{2})\varepsilon'_y\right)^{-\frac{1}{2}}\\
&\times\left(-i\rho(\sigma)-i\rho(k_y(\varepsilon_y,\varepsilon'_y))-im_y-\left(n_1+\tfrac{1}{2}\right)\varepsilon_y+\left(n_2+\tfrac{1}{2}\right)\varepsilon'_y\right)^{-\frac{1}{2}}.
\end{aligned}
\tag{B.3}
$$

The $U(1)$ parts of these one-loop determinants are the same as in the case with no flux. For the vector multiplet, the one-loop determinants can then be written as a product over the positive roots of the gauge algebra.

$$
\begin{aligned}
Z^{\text{vec}}_{\varepsilon_x,\varepsilon'_x} = \left(Z^{U(1)\text{vec}}_{\varepsilon_x,\varepsilon'_x}\right)^{r_G}\prod_{\rho\in\Delta_+}&\frac{1}{|\rho(\sigma)+\rho(k_x(\varepsilon_x,\varepsilon'_x))|}\\
&\times\prod_{n_1,n_2\geq 0}\left((\rho(\sigma)+\rho(k_x(\varepsilon_x,\varepsilon'_x)))^2+\left((n_1+1)\varepsilon_x+(n_2+1)\varepsilon'_x\right)^2\right)^{\frac{1}{2}}\\
&\times\left((\rho(\sigma)+\rho(k_x(\varepsilon_x,\varepsilon'_x)))^2+\left(n_1\varepsilon_x+n_2\varepsilon'_x\right)^2\right)^{\frac{1}{2}},\\
Z^{\text{vec}}_{\varepsilon_y,\varepsilon'_y} = \left(Z^{U(1)\text{vec}}_{-\varepsilon_y,\varepsilon'_y}\right)^{r_G}\prod_{\rho\in\Delta_+}&\frac{1}{|\rho(\sigma)+\rho(k_y(\varepsilon_y,\varepsilon'_y))|}\\
&\times\prod_{n_1,n_2\geq 0}\left((\rho(\sigma)+\rho(k_y(\varepsilon_y,\varepsilon'_y)))^2+\left(-(n_1+1)\varepsilon_y+(n_2+1)\varepsilon'_y\right)^2\right)^{\frac{1}{2}}\\
&\times\left((\rho(\sigma)+\rho(k_y(\varepsilon_y,\varepsilon'_y)))^2+\left(-n_1\varepsilon_y+n_2\varepsilon'_y\right)^2\right)^{\frac{1}{2}}.
\end{aligned}
\tag{B.4}
$$

Similarly we can rewrite the hypermuliplet determinant with a product over the non-trivial weights $\mathbf{R}\setminus\mathbf{R}_0$ of the representation

$$
\begin{aligned}
Z^{\text{hyp}}_{\varepsilon_x,\varepsilon'_x}(\sigma, m_x) = \left(Z^{U(1)\text{hyp}}_{\varepsilon_x,\varepsilon'_x}(m_x)\right)^{|\mathbf{R}_0|}&\\
\times\prod_{\rho\in\mathbf{R}\setminus\mathbf{R}_0}\prod_{n_1,n_2\geq 0}&\left((\rho(\sigma)+\rho(k_x(\varepsilon_x,\varepsilon'_x))+m_x)^2+\left(n_1\varepsilon_x+n_2\varepsilon'_x+\frac{\varepsilon_x+\varepsilon'_x}{2}\right)^2\right)^{-\frac{1}{2}},\\
Z^{\text{hyp}}_{\varepsilon_y,\varepsilon'_y}(\sigma, m) = \left(Z^{U(1)\text{hyp}}_{-\varepsilon_y,\varepsilon'_y}(m_y)\right)^{|\mathbf{R}_0|}&\\
\times\prod_{\rho\in\mathbf{R}\setminus\mathbf{R}_0}\prod_{n_1,n_2\geq 0}&\left((\rho(\sigma)+\rho(k_y(\varepsilon_y,\varepsilon'_y))+m_y)^2+\left(-n_1\varepsilon_x+n_2\varepsilon'_x+\frac{-\varepsilon_x+\varepsilon'_x}{2}\right)^2\right)^{-\frac{1}{2}}.
\end{aligned}
\tag{B.5}
$$

For an adjoint hypermultiplet, the one-loop determinants become

$$
Z^{\mathrm{hyp}}_{\varepsilon_x,\varepsilon'_x}(\sigma,m_x)=\left(Z^{U(1)\mathrm{hyp}}_{\varepsilon_x,\varepsilon'_x}(m_x)\right)^{r_G}\prod_{\rho\in\Delta_+}
$$

$$
\times\prod_{n_1,n_2\geq 0}\left((\rho(\sigma)+\rho(k_x(\varepsilon_x,\varepsilon'_x)))^2+\left(n_1\varepsilon_x+n_2\varepsilon'_x+\frac{\varepsilon_x+\varepsilon'_x}{2}+im_x\right)^2\right)^{-\frac{1}{2}}
$$

$$
\times\left((\rho(\sigma)+\rho(k_x(\varepsilon_x,\varepsilon'_x)))^2+\left(n_1\varepsilon_x+n_2\varepsilon'_x+\frac{\varepsilon_x+\varepsilon'_x}{2}-im_x\right)^2\right)^{-\frac{1}{2}},
$$

$$
Z^{\mathrm{hyp}}_{\varepsilon_x,\varepsilon'_x}(\sigma,\overline{m}_x)=\left(Z^{U(1)\mathrm{hyp}}_{-\varepsilon_x,\varepsilon'_x}(\overline{m}_x)\right)^{r_G}
$$

$$
\times\prod_{\rho\in\Delta_+}\prod_{n_1,n_2\geq 0}\left((\rho(\sigma)+\rho(k_x(\varepsilon_x,\varepsilon'_x)))^2+\left(-n_1\varepsilon_x+n_2\varepsilon'_x+\frac{-\varepsilon_x+\varepsilon'_x}{2}+i\overline{m}_x\right)^2\right)^{-\frac{1}{2}}
$$

$$
\times\left((\rho(\sigma)+\rho(k_x(\varepsilon_x,\varepsilon'_x)))^2+\left(-n_1\varepsilon_x+n_2\varepsilon'_x+\frac{-\varepsilon_x+\varepsilon'_x}{2}-i\overline{m}_x\right)^2\right)^{-\frac{1}{2}}.
$$
(B.6)

From these expressions it is clear that we find the simplifications at the same mass values as in the case without fluxes. Specifically tuning the mass at all fixed points to the the Marcus point we find that the partition function takes the from

$$
\mathcal{Z}_{\mathrm{Marcus}}=\sum_{\{k_i\}}e^{-2\pi i\tau N(\{k_i\})}\int d^{r_G}\sigma\left(\prod_{\rho\in\Delta_+}|\rho(\sigma)|^2\right)\exp\left(-\frac{16\pi^2 s_-}{g_{\mathrm{YM}}^2}\mathrm{Tr}(\sigma^2)\right),
$$
(B.7)

where the sum is over all possible non-trivial values of the flux. Similarly at the Vafa-Witten point we find

$$
\mathcal{Z}=\sum_{\{k_i\}}e^{-2\pi i\tau N(\{k_i\})}\left(Z^G_{\mathrm{V.W}}\right)^{n_+}\left(\overline{Z}^G_{\mathrm{V.W}}\right)^{n_-}
$$
$$
\times\int d^{r_G}\sigma\,\exp\left(-\frac{16\pi^2 s_-}{g_{\mathrm{YM}}^2}\mathrm{Tr}(\sigma^2)\right)\prod_{\rho\in\Delta_+}|\rho(\sigma)|^2\prod_{i\in\{f.p.\}}|\rho(\sigma)+k_i(\varepsilon_i,\varepsilon'_i)|^{-1},
$$
(B.8)

with the last product being over all the fixed points.

## C  Tuning masses on four-manifolds

Here we give a formal proof that at every fixed point of the Killing vector we can tune the mass to whatever value we want.

Pick a fixed point $x_*$ of $v$ and a small neighborhood $U$ of $x_*$ such that the exponential map $\exp:\exp^{-1}(U)\subset T_{x_*}M\to U$ is bijective. Assuming $x_*$ is isolated, there exists $\epsilon>0$ such that $B_\epsilon(0)\subset\exp^{-1}(U)$ and for all $Y\in S^3\subset T_{x_*}M$ the map

$$
f_Y:[0,\epsilon)\to M
$$
$$
r\mapsto\|v(\exp(rY))\|^2,
$$

is strictly monotonically increasing. (This means that if we start going away from $x_*$ the norm of $v$ has to increase.) As a consequence, there exists an $\epsilon'>0$ such that

$$
u:[0,\epsilon')\times S^3\to M
$$
$$
(r,Y)\mapsto\exp\left(f_Y^{-1}(r)Y\right),
$$

parametrizes its image. It holds that $\|v(u(r,Y))\|^2 = r$. Combining this with the fact that $v$ is a Killing vector, we see then that $v(u(r,y)) \in T_{u(r,Y)}u(r,S^3)$.

Let's assume that we have already put some masses $m, \overline{m}$ on $M$, that they satisfy the constraint of being constant along the flow of $v$ but that they do not necessarily have the desired values $m_*, \overline{m}_*$ at the fixed point $x_*$. Pick $0 < \epsilon'' < \epsilon'$ and a function $h : [0, \epsilon''] \to [0,1]$ such that $h(0) = 0, h(\epsilon'') = 1$ and that allows a smooth extension to constant 1 above $\epsilon''$ and constant 0 below 0 (i.e. h is a smooth step function). Define $\pi : \mathbb{R} \times S^3 \to \mathbb{R} : (r,Y) \mapsto r$. With all these inputs we can then define for $p \in M$

$$m'(p) = \begin{cases} h(\pi(u^{-1}(p)))m(p) + (1 - h(\pi(u^{-1}(p)))m_*, \text{ if } p \in u([0,\epsilon''] \times S^3), \\ m(p), \text{ else}, \end{cases}$$

$$\overline{m}'(p) = \begin{cases} h(\pi(u^{-1}(p)))\overline{m}(p) + (1 - h(\pi(u^{-1}(p)))\overline{m}_*, \text{ if } p \in u([0,\epsilon''] \times S^3), \\ \overline{m}(p), \text{ else}. \end{cases}$$

Note that $h \circ \pi \circ u^{-1}$ varies only in a direction transverse to $v$ by construction of $u$ and therefore $m', \overline{m}'$ are constant along the flow of $v$. Also they are smooth due to the constraints on $h$. Finally we have that $m'(x_*) = m_*, \overline{m}'(x_*) = \overline{m}_*$ as desired.

After we have engineered $m, \overline{m}$ the other background fields, $G_\mu$ and $D_{ij}$ and those that depend on them, have to be adjusted.

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
