# Peer review of "Flavor deformations and supersymmetry enhancement in $4d$ $\mathcal{N}=2$ theories"

_SciPost Physics, doi:SciPost Phys. 13, 058 (2022)_

## Round 2 · Referee Report · Anonymous · 2022-1-18

Report

The paper under review is about four-dimensional N=2 superconformal theories placed on general backgrounds in presence of a flavor deformation. The main techniques are introduced at the first stages of the manuscript, in particular the prescription to place an extended supersymmetric theory on a curved manifold as well as the localized partition function on a general background.
The flavor deformations are implemented with a similar method, namely by coupling the flavor current multiplet to a background vector multiplet and by imposing the supersymmetry variation of the fermions to vanish. The authors concentrate on a specific matter content, the N=2* theory with a single adjoint hypermultiplet, where such mechanism generates a dynamical mass term for the hypermultiplet.
The main goal is then to identify specific values of the hypermultiplet mass where the localized partition function extremely simplifies. This method generalizes some previous results (mainly obtained on a four dimensional squashed sphere) to more general backgrounds. Following similar ideas from the three-dimensional set up, the authors show that the simplification of the free energy at special values of the masses corresponds to a supersymmetry enhancement mechanism, proving that such values correspond to some well known twists of N=4 SYM.
The paper is clearly written and shows a good attention to the details. Using the present techniques some previous results in literature can be clarified and extended. Also, in the last section some further applications are discussed, opening new directions both in AGT context and for integrated correlators in N=4 SYM.
For these reasons, I recommend the paper under review for publication in SciPost.

Requested changes

1. At the beginning of section 3.3.1 I suggest to highlight (and clarify) the values for the mass $m_x$ and the definition of $m^*_{\epsilon_x,\epsilon'_x}$ (as well as $m^*_{\epsilon_y,\epsilon'_y}$), since there are frequent references to these values in the following.
2. Typos in the second line of formulas (2.18) and (3.10): I guess $\epsilon_x$ ($\epsilon'_x$) should be replaced by $\epsilon_y$ ($\epsilon'_y$)
3. Small typo in text: "respectivley" slightly before eq. (3.21)
4. Small typo in text: "paramters" 6 lines before the end of pag. 19

---

## Round 2 · Referee Report · Anonymous · 2022-2-1

Report

This paper studies partition functions in 4-dimensional N=2 supersymmetric theories with flavour deformations. The partition functions are computed using supersymmetric localization. In particular, it is found that when the deformation parameters are tuned to special values at fixed points, the partition functions simplify considerably. The underlying mechanism for the simplicity is the enhancement of supersymmetry.

The results help to clarify and generalize the recently observed phenomenon that N=2 partition functions simplify when some parameters take special values. Using the AGT correspondence, the authors comment on the applications of the results to certain one-point functions in Liouville and Toda CFTs. Finally, the construction was applied to study partially integrated correlators in N=4 SYM, generalizing recent developments computing integrated correlators in N=4 SYM using supersymmetric localization. I think the new result from this current paper on the correlators in N=4 SYM still requires further development to make it useful in practice. In particular, one would like to express the constraints in terms of a single function T(U, V), as the authors also commented.

The paper is very well-written and clear, and as I summarized above the results of the paper are interesting and novel, so I suggest the paper be published in SciPost. I found a few punctuation typos:
The period of the sentence right before (3.4) should be removed or be comma?

Similarly
2. The period of Equation (3.18).
3. In equation (4.3), the period after \theta should be moved outside the big braket .
4. The period of Equation (4.11).

---

## Round 3 · Author Response

We thank the referees for their careful reading of the manuscript and their suggestions. We have incorporated all their suggestions in the new manuscript and the list of changes is given.

---

## Round 3 · List of Changes

1. Fixed various punctuations typos in eqs. (3.4), (3.19), (4.3), (4.11) as pointed out by referee 2
2. Highlighted in equation (3.11) the special values for $m_x$ as well as the definition of $m^*_{\varepsilon_x,\varepsilon'_x}$.
3. Replaced $\varepsilon_y$ in eqs. (2.18) and (3.10) by $\varespilon_y$ in eq. (2.18), (3.10).
4. Fixed typos pointed out by referee 1 in eq. 3.21 and on page 19.

---

## Editorial Decision

published